PREPARED FOR SUBMISSION TO JHEP

# Non-hyperbolic 3-manifolds and 3D field theories for 2D Virasoro minimal models

**Dongmin Gang, Heesu Kang, Seongmin Kim**

*Department of Physics and Astronomy & Center for Theoretical Physics,*
*Seoul National University, 1 Gwanak-ro, Seoul 08826, Korea*

*E-mail:* arima275@snu.ac.kr, heesu0434@snu.ac.kr, seongmin0708@snu.ac.kr

ABSTRACT: Using 3D-3D correspondence, we construct 3D dual bulk field theories for general Virasoro minimal models $M(P,Q)$. These theories correspond to Seifert fiber spaces $S^2((P, P-R), (Q, S), (3, 1))$ with two integers $(R, S)$ satisfying $PS - QR = 1$. In the unitary case, where $|P - Q| = 1$, the bulk theory has a mass gap and flows to a unitary topological field theory (TQFT) in the IR, which is expected to support the chiral Virasoro minimal model at the boundary under an appropriate boundary condition. For the non-unitary case, where $|P - Q| > 1$, the bulk theory flows to a 3D $\mathcal{N} = 4$ rank-0 superconformal field theory, whose topologically twisted theory supports the chiral minimal model at the boundary. We also provide a concrete field theory description of the 3D bulk theory using $T[SU(2)]$ theories. Our proposals are supported by various consistency checks using 3D-3D relations and direct computations of various partition functions.

## 1 Introduction

Chiral algebras, also known as vertex algebras, are a powerful and versatile tool in modern theoretical physics. They provide a unifying algebraic framework for understanding diverse physical phenomena, ranging from two-dimensional conformal field theories to the intricate structures in higher dimensional ($D \geq 3$) topological or supersymmetric quantum field theories [1–7]. Among chiral algebras, rational chiral algebras have rich and rigid mathematical structures and broad applications in physics. They allow only a finite number of irreducible representations, whose characters form vector-valued modular forms. They are closely related to 3D topological field theories via the so-called bulk-boundary correspondence, which provides knot or 3-manifold invariants and describes the universal behaviors of topological phases. The rigid mathematical structures greatly simplify the classification program of rational chiral algebras, and several important classes have been classified. The most famous and successful class is the Virasoro minimal models $M(P,Q)$ [8], which describe universal features of critical phenomena in various 2D systems, such as the Ising model.

In this paper, we study the 3D bulk theories related to the Virasoro minimal models via the bulk-boundary correspondence. In the case of unitary rational chiral algebras, the bulk theories have a mass gap, and the infrared (IR) physics is described by unitary topological field theories. The IR topological quantum field theories (TQFTs) share common modular tensor category structures with the corresponding boundary rational chiral algebras [9–11].

For non-unitary rational chiral algebras, recent studies have shown that the bulk theories can be described by topologically twisted theories of an exotic class of superconformal field theories (SCFTs) called 3D $\mathcal{N} = 4$ rank-0 SCFTs [12–20]. Here, 'rank-0' denotes the absence of Coulomb and Higgs branch operators in the theory. This exotic property proves crucial in realizing rational chiral algebra at the boundary [17, 21, 22]. Our approach begins by realizing the 3D bulk theories for minimal models $M(P,Q)$ through the 3D-3D correspondence [23–25] with Seifert fiber spaces, as depicted below:

$$\boxed{\text{Seifert fiber spaces}} \xrightarrow{\text{3D}-\text{3D}} \boxed{\text{3D theories (gapped theories or } \mathcal{N} = 4 \text{ rank-0 SCFTs)}}$$
$$\xrightarrow{\text{bulk-boundary}} \boxed{\text{2D (unitary or non-unitary) minimal models}}$$

$$(1.1)$$

For details of the proposal, refer to (2.10) and (2.24). Furthermore, we provide a field-theoretic depiction (3.22) of these 3D theories by utilizing the topological structures of the Seifert fiber spaces. Our construction provides a unified framework for the bulk duals of both unitary and non-unitary minimal models.

The rest of this paper is organized as follows. In Section 2, we introduce the Virasoro minimal model $M(P,Q)$ and its 3D dual theory $\mathcal{T}_{(P,Q)}$, along with the basic dictionaries of the bulk-boundary correspondence. We then propose that the theories $\mathcal{T}_{(P,Q)}$ can be realized as 3D class R theories associated with Seifert fiber spaces (see (2.24)). This proposal is tested against various non-trivial 3D-3D relations and bulk-boundary dictionaries, as summarized in Table 1. In Section 3, we provide an explicit field theory description for $\mathcal{T}_{(P,Q)}$ (see (3.22)) along with several non-trivial consistency checks. The Appendices contain technical details of the supersymmetric partition function computations and the identification of decoupled topological field theories from 1-form symmetry 't Hooft anomalies.

## 2  Virasoro minimal models from 3-manifolds

In this section, we begin by reviewing basic aspects of Virasoro minimal models. Then, we introduce the bulk-boundary correspondence and the 3D-3D correspondence, summarized in Table 1. Utilizing these correspondences as guidelines, we propose the 3D bulk duals of minimal models as 3D class R theories associated with 3-manifolds known as Seifert fiber spaces, as given in (2.24).

### 2.1  Virasoro minimal model $M(P,Q)$

The minimal model $M(P,Q) = M(Q,P)$ is labeled by two integers, $P$ and $Q$, subject to the following conditions:

$$P, Q \geq 2 \text{ and } \gcd(P,Q) = 1 . \tag{2.1}$$

The underlying chiral algebra is the Virasoro algebra, with the 2D central charge given by:

$$c_{2d} = 1 - \frac{6(P-Q)^2}{PQ} . \tag{2.2}$$

The model includes critical Ising CFT $M(3,4)$, tricritical Ising CFT $M(4,5)$ and Lee-Yang CFT $M(2,5)$. The 2D RCFT can be unitary or non-unitary, depending on $(P,Q)$:

$$M(P,Q) \text{ is } \begin{cases} \text{unitary,} & \text{if } |P-Q| = 1 \\ \text{non-unitary,} & \text{otherwise .} \end{cases} \tag{2.3}$$

There are $N_{(P,Q)} := \frac{(P-1)(Q-1)}{2}$ primaries $\mathcal{O}_{(a,b)}$ labeled by two integers $1 \le a < P$ and $1 \le b < Q$ modulo an equivalence relation $\mathcal{O}_{(a,b)} = \mathcal{O}_{(P-a,Q-b)}$. The conformal dimensions $h$ of the primaries are given by:

$$h_{(a,b)} = \frac{(Pb - Qa)^2 - (P-Q)^2}{4PQ} , \tag{2.4}$$

and the conformal characters are:

$$\chi_{(a,b)}(q) = \frac{q^{h_{(a,b)} - \frac{c_{2d}}{24}}}{(q)_\infty} \sum_{n \in \mathbb{Z}} \left( q^{n^2 PQ + n(Qa - Pb)} - q^{(nP+a)(nQ+b)} \right) . \tag{2.5}$$

Here $(q)_\infty = \prod_{k=1}^\infty (1 - q^k)$ as usual. Under the S-transformation, $q := e^{2\pi i \tau} \to \tilde{q} := e^{2\pi i (-\frac{1}{\tau})}$, the characters transforms as:

$$\chi_\alpha(\tilde{q}) = \sum_\beta S_{\alpha\beta} \chi_\beta(q) , \tag{2.6}$$

with the modular $S$-matrix given by:

$$S_{(a,b),(a',b')} = -\sqrt{\frac{8}{PQ}} (-1)^{ba' + b'a} \sin(\pi \frac{P}{Q} bb') \sin(\pi \frac{Q}{P} aa') . \tag{2.7}$$

In the above, $\alpha, \beta = 0, 1, \ldots, N_{(P,Q)} - 1$ are collective indices for the primaries, where $\alpha = 0$ corresponds to the vacuum module, i.e., $\alpha = 0 \leftrightarrow (a,b) = (1,1)$.

## 2.2 Bulk dual 3D $\mathcal{T}_{(P,Q)}$ theory

The bulk-boundary correspondence relates a 3D bulk (semi-simple and finite) topological field theory $\mathcal{T}$ to 2D chiral rational conformal field theories (RCFTs). The boundary chiral RCFT $\chi R$ depends on the choice of holomorphic boundary condition $\mathbb{B}$, and the corresponding RCFT will be denoted as $\chi R[\mathcal{T}, \mathbb{B}]$:

$$\text{(3D topological field theory } \mathcal{T}) \xrightarrow{\text{at boundary with } \mathbb{B}} \text{(2D chiral RCFT } \chi R[\mathcal{T}, \mathbb{B}]) . \tag{2.8}$$

To realize the full (diagonal) RCFT $\mathcal{R}$, one needs to put the bulk theory on an interval, $\mathcal{M}_2 \times [0,1]$, with the holomorphic boundary condition $\mathbb{B}$ on both boundaries. In the IR, the system flows to the 2D RCFT $\mathcal{R}[\mathcal{T}, \mathbb{B}]$ on $\mathcal{M}_2$.

For unitary case, the bulk theory is a unitary TQFT, which describes the universal IR behavior of a (2+1)D gapped system, such as fractional quantum Hall system. For non-unitary case, recent studies show that the bulk theories can be described by topologically twisted theories of an exotic class of superconformal field theories called 3D $\mathcal{N} = 4$ rank-0

SCFTs. The bulk-boundary correspondence for the non-unitary case can be summarized as follows:

$$\text{a 3D } \mathcal{N} = 4 \text{ rank-0 SCFT } \mathcal{T} \xrightarrow{\text{a top'l twisting}} \text{non-unitary TQFT } \mathcal{T}^{\text{top}}$$

$$\xrightarrow{\text{at boundary with } \mathbb{B}} \text{non-unitary chiral RCFT } \chi \mathcal{R}[\mathcal{T}, \mathbb{B}] \,. \tag{2.9}$$

Rank-0 means there is no Coulomb and Higgs branch operators in the theory. The exotic property turns out to be crucial to realize rational chiral algebra at the boundary after a topological twisting. There are two possible choices of topological twistings ('top'=$A$ or $B$) denoted as $A$ and $B$ twistings.

Let $\mathcal{T}_{(P,Q)}$ be the bulk dual theory related to the chiral minimal model $\chi M(P,Q)$ via the bulk-boundary correspondence:

**Def : 3D $\mathcal{T}_{(P,Q)}$ theory is defined as**

For $|P - Q| = 1$, $\mathcal{T}_{(P,Q)}$ is a 3D unitary TQFT with

$$\mathcal{T}_{(P,Q)} \xrightarrow{\text{at boundary with a proper } \mathbb{B}} \chi M(P,Q) \,,$$

For $|P - Q| > 1$, $\mathcal{T}_{(P,Q)}$ is a $\mathcal{N} = 4$ 3D rank-0 SCFT with

$$\mathcal{T}_{(P,Q)} \xrightarrow{\text{a top'l twisting}} \text{non-unitary TQFT } \mathcal{T}^{\text{top}}_{(P,Q)} \xrightarrow{\text{at boundary with a proper } \mathbb{B}} \chi M(P,Q) \,. \tag{2.10}$$

The main goal of this paper is to construct the bulk theory $\mathcal{T}_{(P,Q)}$ dual to the minimal model $M(P,Q)$.

Basic dictionaries of the bulk-boundary correspondence are summarized in the first and second column of Table 1. Refer to [14, 16] for details. In the table $T_{\text{irred}}[M]$ corresponds to the bulk theory $\mathcal{T}$. We will realize the bulk theory as an IR fixed point of 3D $\mathcal{N} \geq 2$ supersymmetric gauge theories, and the bulk quantities in the table are related to the partition functions on various symmetric backgrounds, which are RG-invariant. When the SUSY background is $\mathcal{M}_{g,p}$, a degree $p$ circle bundle over a genus $g$ Riemann surface, the partition function can be given in the following form [26–29]

$$\mathcal{Z}^{\mathcal{M}_{g,p}} = \sum_{\vec{x}_\alpha \in \mathcal{S}_{\text{BE}}} \mathcal{H}(\vec{x}_\alpha)^{g-1} \mathcal{F}(\vec{x}_\alpha)^p \,. \tag{2.11}$$

Here $\vec{x}_\alpha \in \mathcal{S}_{\text{BE}}$ are called Bethe-vacua, which are ground states on a two-torus $\mathbb{T}^2$ when the bulk theory is a topological field theory. $\mathcal{H}$ and $\mathcal{F}$ are called 'handle-gluing' and 'fibering' operators, respectively. Let $\mathcal{Z}^{S^3_b}$ denotes the supersymmetric partition function on a squashed 3-sphere $S^3_b$ [30–32]:

$$S^3_b := \{(z, w) \in \mathbb{C}^2 \ : \ b^2 |z|^2 + b^{-2} |w|^2 = 1\} \,. \tag{2.12}$$

When the bulk theory flows to a 3D $\mathcal{N} = 4$ rank-0 SCFT, the SUSY prtition functions have non-trivial dependence on $(M, \nu)$. In terms of an $\mathcal{N} = 2$ subalgebra, the $\mathcal{N} = 4$ theory has a flavor symmetry $U(1)_A$ whose charge $A$ is given by:

$$A = J^C_3 - J^H_3 \,, \tag{2.13}$$

where $J_3^C$ and $J_3^H$ are two Cartans of $SU(2)^C \times SU(2)^H \simeq SO(4)$ R-symmetry of the $\mathcal{N} = 4$ theory. They are normalized as $J_3^{C/H} \in \mathbb{Z}/2$. $M$ denotes the properly rescaled real mass of the $U(1)_A$ symmetry. $\nu$ parametrizes the mixing between the $U(1)_R$ symmetry of the $\mathcal{N} = 2$ subalgebra and the $U(1)_A$ symmetry as follows:

$$R_\nu = R_{\nu=0} + \nu A = \left( J_3^C + J_H^3 \right) + \nu \left( J_3^C - J_H^3 \right) . \tag{2.14}$$

For rank-0 SCFTs, the supersymmetric partition functions in the following limits:

$$\begin{aligned} \text{A-twisting limit} &: (M, \nu) \to (0, \nu = -1) , \\ \text{B-twisting limit} &: (M, \nu) \to (0, \nu = 1) , \end{aligned} \tag{2.15}$$

are known to compute the partition functions of topologically A-twisted or B-twisted theory, respectively. In the limits, the squashed 3-sphere function becomes independent on the squashing parameter modulo a phase factor in (A.2). In the table, we define

$$\begin{aligned} \{ \mathcal{F}^A, \mathcal{H}^A, \mathcal{Z}_b^A \} &:= \{ \mathcal{F}, \mathcal{H}, \mathcal{Z}^{S_b^3} \}|_{(M,\nu) \to (0,-1)} , \\ \{ \mathcal{F}^B, \mathcal{H}^B, \mathcal{Z}_b^B \} &:= \{ \mathcal{F}, \mathcal{H}, \mathcal{Z}^{S_b^3} \}|_{(M,\nu) \to (0,+1)} . \end{aligned} \tag{2.16}$$

On the hand, the SUSY partition functions of a 3D $\mathcal{N} = 4$ rank-0 SCFT at $(M = 0, \nu = 0)$ computes the partition functions at the superconformal point, and we define:

$$\mathcal{Z}_b^{\text{con}} := \mathcal{Z}^{S_b^3}|_{(M,\nu) \to (0,0)} . \tag{2.17}$$

When the 3D gauge theory has a mass gap and flows to a unitary TQFT in the IR, the SUSY partition functions are independent on the $(M, \nu)$ and its squashed 3-sphere partition function is independent on the $b$ modulo a phase factor in (A.2).

## 2.3 $\mathcal{T}_{(P,Q)}$ from non-hyperbolic 3-manifolds

Let $T_{\text{irred}}[M]$ denote the 3D class R theory associated with a closed 3-manifold $M$, whose field theory description is proposed in [24, 25]. The theory is believed to describe an effective 3D field theory of 6D $A_1$ $\mathcal{N} = (2,0)$ superconformal field theory compactified on the 3-manifold $M$. The subscript 'irred' emphasizes that the theory only see an irreducible component of $SL(2,\mathbb{C})$ flat connections on $M$ rather than all flat connections [33]. The theory should be distinguished from $T_{\text{full}}[M]$ studied in [4, 34–40], which is assumed to see the all flat connections. Unlike $T_{\text{irred}}$, however, there is no known systematic algorithm for constructing the field theory description of $T_{\text{full}}[M]$ for a general 3-manifold $M$. It is even uncertain whether the theory $T_{\text{full}}[M]$ exists as a genuine 3D field theory for general $M$ [25].

For Seifert fiber spaces (SFSs) $M = S^2(\vec{p}, \vec{q})$ in Figure 1, the IR phases of the 3D field theory $T_{\text{irred}}[M]$ have been analyzed in [41]. It was empirically found that

$$T_{\text{irred}}[M = S^2((p_1, q_1), (p_2, q_2), (p_3, q_3))]$$

$$\xrightarrow{\text{In the IR}} \begin{cases} \text{an unitary TQFT,} & q_i = \pm 1 \pmod{p_i} \ \forall i = 1, 2, 3 , \\ \text{a rank-0 SCFT,} & \text{otherwise} \end{cases} \tag{2.18}$$

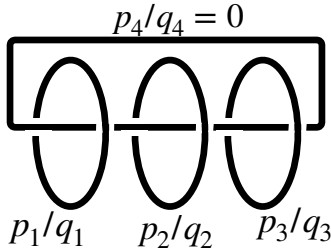

**Figure 1**. A Dehn surgery representation of $S^2(\vec{p}, \vec{q}) := S^2((p_1, q_1), (p_2, q_2), (p_3, q_3))$.

By combining the 3D-3D correspondence for the Seifert fiber spaces and the bulk-boundary correspondence, one can consider following correspondence:

$$M = S^2(\vec{p}, \vec{q}) \xrightarrow{\text{3D-3D correspondence}} T_{\text{irred}}[M] \xrightarrow{\text{Bulk-boundary}} \text{2D chiral RCFT } \chi\mathcal{R}[M; \mathbb{B}] .$$

(2.19)

Basic dictionaries for the correspondence are summarized in Table 1. Let us briefly explain

| 2D chiral RCFT $\chi\mathcal{R}[M; \mathbb{B}]$ | $T_{\text{irred}}[M]$ | $SL(2, \mathbb{C})$ CS on $M$ |
|---|---|---|
| (unitary)/(non-unitary) | (mass gap)/(rank-0 SCFT) | equation (2.23) |
| Primary $\mathcal{O}_{\alpha=0,\dots,N-1}$ | Bethe-vacuum $\vec{x}_\alpha \in \mathcal{S}_{\text{BE}}$ | $\rho_\alpha \in \chi_{\text{irred}}[M]$ |
| Conformal dimension $h_\alpha$ | $e^{2\pi i h_\alpha} = \mathcal{F}^{\text{top}}(\vec{x}_\alpha)/\mathcal{F}^{\text{top}}(\vec{x}_{\alpha=0})$ | $e^{2\pi i h_\alpha} = e^{2\pi i (\text{CS}[\rho_{\alpha=0}] - \text{CS}[\rho_\alpha])}$ |
| $S_{0\alpha}^2$ | $(\mathcal{H}^{\text{top}}(\vec{x}_\alpha))^{-1}$ | $1/(2\text{Tor}[\rho_\alpha])$ |
| $\lvert S_{00} \rvert$ | $\lvert \mathcal{Z}_b^{\text{top}} \rvert$ | $\left\lvert \sum_{\rho_\alpha} \frac{e^{-2\pi i \text{CS}[\rho_\alpha]}}{2\text{Tor}[\rho_\alpha]} \right\rvert$ |
| $\min_\alpha\{\lvert S_{0\alpha} \rvert\}$ | $e^{-F} = \lvert \mathcal{Z}_{b=1}^{\text{con}} \rvert$ | $\min_\alpha\{1/\sqrt{2\text{Tor}[\rho_\alpha]}\}$ |

**Table 1**. Basic dictionaries for the correspondence among non-hyperbolic 3-manifolds $M$, 3D bulk theories $T_{\text{irred}}[M]$, and 2D chiral RCFTs $\chi\mathcal{R}[M]$ for $M = S^2((\vec{p}, \vec{q}))$ with $H^1(M, \mathbb{Z}_2) = 1$. The superscripts 'top'$=A$ (or $B$) and 'con' denote the partition function in $A$ (or $B$)-twisting limit (2.15) and at the superconformal point (2.17) of rank-0 SCFT respectively. For the mass gap case, the superscripts can be ignored. The $S_{\alpha\beta}$ is the modular S-matrix in (2.6) and (2.7). $\mathcal{H}$ and $\mathcal{F}$ are handle gluing and fibering operators appearing in the twisted partition function (2.11). $\mathcal{Z}_b$ is the squashed 3-sphere partition function. $\text{CS}[\rho]$ and $\text{Tor}[\rho]$ denotes the Chern-Simons invariant and adjoint Reidemeister torsion of an irreducible flat connection $\rho$.

the basic 3D-3D dictionaries in the table. Refer to [42–47] for details. The dictionary is valid only for the 3-manifold with trivial $H^1(M, \mathbb{Z}_2)$. The $\chi_{\text{irred}}(M)$ in the table is the set of (adjoint-)irreducible $SL(2, \mathbb{C})$ characters on a 3-manifold $M$, which is defined as:

$$\chi_{\text{irred}}[M] = \{\rho \in \text{Hom}\,[\pi_1(M) \to SL(2, \mathbb{C})] \,:\, \dim H(\rho) = 0\}/\sim ,$$
$$\text{where } H(\rho) := \{g \in SL(2, \mathbb{C}) \,:\, [g, \rho(a)] = 0 \ \forall a \in \pi_1(M)\} .$$

(2.20)

The equivalence relation is defined as[1]

$$\rho_1 \sim \rho_2 \text{ if } \text{Tr}(\rho_1(a)) = \text{Tr}(\rho_2(a)) , \ \ \forall a \in \pi_1(M) .$$

(2.21)

---

[1]In [45], they consider equivalence up to $SL(2, \mathbb{C})$ conjugation. But the trace equivalence relation $\sim$, which is a weaker equivalence, is turned out be more relevant in the 3D-3D correspondence [46].

The condition $\dim H(\rho) = 0$ corresponds to the irreducibility of the homomorphism $\rho$. A homomorphism $\rho$ defines an $SL(2,\mathbb{C})$ flat connection $\mathcal{A}_\rho$, $d\mathcal{A}_\rho + \mathcal{A}_\rho \wedge \mathcal{A}_\rho = 0$. The CS$[\rho]$ and Tor$[\rho]$ are basic topological invariants of the flat connection called Chern-Simons invariant and the adjoint Reidemeister torsion, respectively. The CS invariant is defined as

$$\mathrm{CS}[\rho] := \frac{1}{8\pi^2} \int \mathrm{Tr}\left(\mathcal{A}_\rho d\mathcal{A}_\rho + \frac{2}{3}\mathcal{A}_\rho^3\right) \pmod 1 . \tag{2.22}$$

The adjoint torsion Tor$[\rho]$ appears as the 1-loop part of perturbative expansion of $SL(2,\mathbb{C})$ CS theory around the flat connection $\mathcal{A}_\rho$ [9]. For $M = S^2(\vec{p},\vec{q})$ with trivial $H^1(M,\mathbb{Z}_2)$, one can determine the IR phase of the $T_{\mathrm{irred}}[M]$ theory in the following way [41, 45]

$$T_{\mathrm{irred}}[M = S^2(\vec{p},\vec{q})]$$

$$\xrightarrow{\text{in the IR}} \begin{cases} \text{unitary TQFT} , & \text{if } |\sum_{\rho\in\chi} \frac{e^{-2\pi i \mathrm{CS}[\rho_\alpha]}}{2\mathrm{Tor}[\rho]}| \leq \frac{1}{\sqrt{|2\mathrm{Tor}[\rho_\alpha]|}} \ \forall \rho_\alpha \in \chi \\ \text{3D rank-0 SCFT} , & \text{otherwise} \end{cases} \tag{2.23}$$

Here, $\chi$ abbreviates $\chi_{\mathrm{irred}}[M]$.

**As the main result of this section**, we propose that

$$\mathcal{T}_{(P,Q)} \simeq T_{\mathrm{irred}}\left[S^2((P,P-R),(Q,S),(3,1))\right] . \tag{2.24}$$

The 3D theory $\mathcal{T}_{(P,Q)}$ is defined as the bulk dual of the Virasoro minimal model $M(P,Q)$ as in (2.10). Here $(R,S)$ is chosen to satisfy

$$\begin{pmatrix} P & Q \\ R & S \end{pmatrix} \in SL(2,\mathbb{Z}) . \tag{2.25}$$

It fixes the $(R,S)$ modulo a shift $(R,S) \to (R,S) + \mathbb{Z}(P,Q)$, and we will claim that the $T_{\mathrm{irred}}\left[S^2((P,R-P),(Q,-S),(3,1))\right]$ is independent of the shift:

$$T_{\mathrm{irred}}\left[S^2((P,P-R),(Q,S),(3,1))\right] \simeq T_{\mathrm{irred}}\left[S^2((P,P-\tilde{R}),(Q,\tilde{S}),(3,1))\right]$$
$$\text{where } (\tilde{R},\tilde{S}) = (R,S) + n(P,Q) \text{ and for arbitrary } n \in \mathbb{Z} . \tag{2.26}$$

Throughout this paper, we use the following two equivalence relations denoted by $\sim$ and $\simeq$ among 3D gauge theories,

$\mathcal{T}_1 \sim \mathcal{T}_2$ if two theories are IR equivalent up to some 'topological operations' ,

$\mathcal{T}_1 \simeq \mathcal{T}_2$ if two theories are IR equivalent up to some 'minimal topological operations' .

$$\tag{2.27}$$

Topological operations include tensoring with a unitary TQFT, gauging of finite (generalized) symmetries, time-reversal and so on. On the other hand, the minimal topological operations are topological operations which preserve the absolute values of partition functions on arbitrary closed 3-manifolds. The minimal ones include tensoring with an invertible

TQFT, time-reversal and so on. Notice that $\simeq$ is a stronger equivalence than $\sim$. The $\mathcal{T}_{(P,Q)}$, like $M(P,Q)$, is invariant under the exchange of $P \leftrightarrow Q$:

$$
\begin{aligned}
\mathcal{T}_{(Q,P)} &\simeq T_{\text{irred}}\left[S^2((Q,Q-\tilde{R}),(P,\tilde{S}),(3,1))\right] \text{ with } \begin{pmatrix} Q & P \\ \tilde{R} & \tilde{S} \end{pmatrix} \in SL(2,\mathbb{Z}) \, . \\
&\simeq T_{\text{irred}}\left[S^2((P,(\tilde{S}-P)+P),(Q,(Q-\tilde{R})),(3,1))\right] \simeq \mathcal{T}_{(P,Q)} \, .
\end{aligned}
\tag{2.28}
$$

In the 2nd line, we use the fact that $\begin{pmatrix} P & Q \\ -\tilde{S}+P & Q-\tilde{R} \end{pmatrix} \in SL(2,\mathbb{Z})$.

Let us check the proposal using the dictionaries in Table 1. First, note that the 3-manifold $S^2((P,P-R),(Q,S),(3,1))$ has trivial $H^1(M,\mathbb{Z}_2)$[2] and thus one can use the dictionaries. The fundamental group of the SFS can be presented as:

$$
\pi_1(S^2(\vec{p},\vec{q})) = \langle x_1, x_2, x_3, h | x_i^{p_i} h^{q_i} = 1, x_1 x_2 x_3 = 1, h \text{ is central} \rangle \, .
\tag{2.29}
$$

As studied in [46], irreducible characters on $M = S^2(\vec{p},\vec{q})$ can be specified by the quadruple $(n_1,n_2,n_3,\lambda)$ with $n_{k=1,2,3} \in \mathbb{Z}_{\geq 0}/2$ and $\lambda \in \{0,\frac{1}{2}\}$, where

$$
\text{Tr}(\rho(x_k)) = 2\cos(\frac{2\pi n_k}{p_k}) \, , \ \rho(h) = \exp(2\pi i\lambda)\mathbb{I}_2 \, .
\tag{2.30}
$$

Note that the $\rho(h)$ should be an element of center subgroup $\mathbb{Z}_2$ ($\mathbb{I}_2$ or $-\mathbb{I}_2$) of $SL(2,\mathbb{C})$ in order for $\rho$ to be (adjoint)-irreducible, i.e. $\dim H(\rho) = 0$. Otherwise, the fundamental group relation can only be met when $\rho(x_k) \in \{\pm\mathbb{I}_2\}$ for all $k = 1,2,3$, and thus $\dim H(\rho) > 0$.

Before going into the details of the character variety, let us first check the proposed duality in (2.26) at the level of the character variety, which corresponds to the set of Bethe-vacua in the $T_{\text{irred}}[M]$ theory. One can easily construct a natural one-to-one map between the character varieties with different choices of $(R,S)$ as follows

$$
\begin{aligned}
\rho &\in \chi_{\text{irred}}\left[S^2((P,P-R),(Q,S),(3,1))\right] \\
&\xleftarrow{\text{one-to-one}} \tilde{\rho} \in \chi_{\text{irred}}\left[S^2((P,P-\tilde{R}),(Q,\tilde{S}),(3,1))\right]_{(\tilde{R},\tilde{S})=(R,S)+n(P,Q)} \, ,
\end{aligned}
\tag{2.31}
$$

where
$$
\tilde{\rho}(x_1) = \rho(x_1)\rho(h)^n, \ \ \tilde{\rho}(x_2) = \rho(x_2)\rho(h)^{-n}, \ \ \tilde{\rho}(x_3) = \rho(x_3), \ \ \tilde{\rho}(h) = \rho(h) \, .
$$

Basic invariants, CS$[\rho]$ and Tor$[\rho]$, of irredicible characters are preserved under the map.

The character variety is studied in [46] and they found that

$$
\begin{aligned}
\chi_{\text{irred}}(M) = &\left\{ \left( n_{p_1,q_1}(j_1), n_{p_2,q_2}(j_2), n_{p_3,q_3}(j_3), \frac{1}{2} \right) \Big| j_k \in [0,\dots,p_k-2]^e \right\} \\
&\bigsqcup \left\{ (n_{p_1,q_1}(j_1), n_{p_2,q_2}(j_2), n_{p_3,q_3}(j_3), 0) \Big| j_k \in [0,\dots,p_k-2]^o \right\},
\end{aligned}
\tag{2.32}
$$

---

[2] $S^2(\vec{p},\vec{q})$ has trivial $H^1(M,\mathbb{Z}_2)$ if and only if $p_1 p_2 p_3 \left( \frac{q_1}{p_1} + \frac{q_2}{p_2} + \frac{q_3}{p_3} \right) \in 2\mathbb{Z}+1$.

where

$$n_{p,q}(j) = \begin{cases} \frac{p-j-1}{2}\,, & q \text{ and } j \in 2\mathbb{Z} \\ \frac{j+1}{2}\,, & \text{otherwise} \end{cases} \tag{2.33}$$

Here $[0,\ldots,p]^{e/o}$ denotes the set of even/odd numbers between $0$ and $p$. the So the irreducible characters on $S^2(\vec{p},\vec{q})$ are labeled by $\vec{j} \in \prod_{k=1}^3 [0...p_k-2]^e \sqcup \prod_{k=1}^3 [0,...,p_k-2]^o$. In our case, $p_3 = 3$ and $j_3 \in \{0,1\}$, so $(j_1,j_2) \in \prod_{k=1}^2 [0,...,p_k-2]^e \sqcup \prod_{k=1}^2 [0,...,p_k-2]^o$ wholly fix the $\vec{j}$ as well as $\lambda$. In the labeling, the adjoint Reidemeister torsion and Chern-Simons invariant (mod 1) of a $SL(2,\mathbb{C})$ character $\rho_{\vec{j}}$ on the 3-manifold $S^2((P, R-P),(Q,-S),(3,1))$ are [46]

$$\text{Tor}(\rho_{\vec{j}}) = \prod_{k=1}^3 \frac{p_i}{4\sin^2\left(\frac{2\pi r_k n_k}{p_k}\right)} = \frac{PQ}{16\sin^2\left(\frac{2\pi Qn_1}{P}\right)\sin^2\left(\frac{2\pi Pn_2}{Q}\right)}\,,$$

$$\text{CS}(\rho_{\vec{j}}) = \sum_{k=1}^3 \frac{-c_k}{4p_k}(j_k+1)^2\,. \tag{2.34}$$

where $c_k$ is :

$$c_k = \begin{cases} p_k q_k s_k - r_k\,, & q_k \text{ odd} \\ p_k q_k s_k - r_k(p_k-1)^2, & q_k \text{ even} \end{cases} \tag{2.35}$$

Here $(r_k, s_k) \in \mathbb{Z}^2$ is chosen to satisfy $p_k s_k - q_k r_k = 1$, and $n_1 = n_{(p_1,q_1)}(j_1)$ and $n_2 := n_{(p_2,q_2)}(j_2)$ in (2.33).

We propose the following one-to-one map between the primaries $\mathcal{O}_{(a,b)}$ of $M(P,Q)$ and the irreducible characters $\rho_{\vec{j}}$ on $S^2((P,P-R),(Q,S),(3,1))$:

$$\mathcal{O}_{(a,b)} \leftrightarrow \rho_{\vec{j}}\,, \text{ where}$$

$$i)\ (a,b) = \begin{cases} (j_1+1, j_2+1)\,, & j_1, j_2 \text{ even} \\ (P-j_1-1, j_2+1)\,, & j_1, j_2 \text{ odd} \end{cases},$$

$$\text{if } (P,R,S) \in (4\mathbb{Z}, 2\mathbb{Z}+1, 2\mathbb{Z}) \text{ or } (Q,R,S) \in (4\mathbb{Z}, 2\mathbb{Z}+1, 2\mathbb{Z}+1) \tag{2.36}$$

$$ii)\ (a,b) = \begin{cases} (P-j_1-1, j_2+1)\,, & j_1, j_2 \text{ even} \\ (j_1+1, j_2+1)\,, & j_1, j_2 \text{ odd} \end{cases},$$

otherwise .

Under the map, one can check that $(2\text{Tor}(\rho_\alpha))^{-1}$ (resp. $\text{CS}[\rho_{\alpha=0}] - \text{CS}[\rho_\alpha]$) equals to $S_{0\alpha}^2$ (resp. $h_\alpha \pmod 1$) of $M(P,Q)$.

Let us give some concrete examples.

**Example:** $M(3,4) = (\textbf{Ising CFT})$ **from** $S^2((3,4),(4,-1),(3,1))$   We choose $(R,S) = (-1,-1)$. There are 3 irreducible $SL(2,\mathbb{C})$ characters $\rho_{\vec{j}}$:

| $(j_1, j_2)$ | $(\vec{n}, \lambda)$ | $\mathrm{CS}[\rho]$ | $\mathrm{Tor}[\rho]$ | $\mathcal{O}_{(a,b)}$ | $\mathrm{CS}[\rho_{\alpha=0}] - \mathrm{CS}[\rho]$ |
|---|---|---|---|---|---|
| $(0,0)$ | $(1, \frac{1}{2}, \frac{1}{2}, \frac{1}{2})$ | $\frac{31}{48}$ | $2$ | $\mathcal{O}_{(1,1)} = \mathcal{O}_{(2,3)}$ | $0$ |
| $(0,2)$ | $(1, \frac{3}{2}, \frac{1}{2}, \frac{1}{2})$ | $\frac{7}{48}$ | $2$ | $\mathcal{O}_{(1,3)} = \mathcal{O}_{(2,1)}$ | $\frac{1}{2}$ |
| $(1,1)$ | $(1, 1, 1, 0)$ | $\frac{7}{12}$ | $1$ | $\mathcal{O}_{(1,2)} = \mathcal{O}_{(2,2)}$ | $\frac{1}{16}$ |

**Table 2**. Ising CFT from $S^2((3,4), (4,-1), (3,1))$. We use the characters-to-primaries map in (2.36). The result is compatible with that $S_{(1,1),(1,1)} = S_{(1,1),(2,1)} = \frac{1}{2}, S_{(1,1),(2,2)} = \frac{1}{\sqrt{2}}$ and $h_{(2,1)} = \frac{1}{2}, h_{(2,2)} = \frac{1}{16}$.

**Example:** $M(3,4) = $ (**Ising CFT**) from $S^2((3,1), (4,3), (3,1))$ This time, we choose $(R, S) = (2,3)$. Due to the different choice of $(R, S)$, the correspondence between the primaries and the irreducible characters is different from the case when $(R, S) = (-1, -1)$.:

| $(j_1, j_2)$ | $(\vec{n}, \lambda)$ | $\mathrm{CS}[\rho]$ | $\mathrm{Tor}[\rho]$ | $\mathcal{O}_{(a,b)}$ | $\mathrm{CS}[\rho_{\alpha=0}] - \mathrm{CS}[\rho]$ |
|---|---|---|---|---|---|
| $(0,0)$ | $(\frac{1}{2}, \frac{1}{2}, \frac{1}{2}, \frac{1}{2})$ | $\frac{7}{48}$ | $2$ | $\mathcal{O}_{(2,1)} = \mathcal{O}_{(1,3)}$ | $\frac{1}{2}$ |
| $(0,2)$ | $(\frac{1}{2}, \frac{3}{2}, \frac{1}{2}, \frac{1}{2})$ | $\frac{31}{48}$ | $2$ | $\mathcal{O}_{(2,3)} = \mathcal{O}_{(1,1)}$ | $0$ |
| $(1,1)$ | $(1, 1, 1, 0)$ | $\frac{7}{12}$ | $1$ | $\mathcal{O}_{(2,2)} = \mathcal{O}_{(1,2)}$ | $\frac{1}{16}$ |

**Table 3**. Ising CFT from $S^2((3,1), (4,3), (3,1))$. We use the characters-to-primaries map in (2.36). The result is compatible with that $S_{(1,1),(1,1)} = S_{(1,1),(2,1)} = \frac{1}{2}, S_{(1,1),(2,2)} = \frac{1}{\sqrt{2}}$ and $h_{(2,1)} = \frac{1}{2}, h_{(2,2)} = \frac{1}{16}$.

**Example:** $M(5,2) = $ (**Lee-Yang CFT**) from $S^2((5,3), (2,1), (3,1))$ We choose $(R, S) = (2,1)$. There are 2 irreducible $SL(2, \mathbb{C})$ characters $\rho_{\vec{j}}$:

| $(j_1, j_2)$ | $(\vec{n}, \lambda)$ | $\mathrm{CS}[\rho]$ | $\mathrm{Tor}[\rho]$ | $\mathcal{O}_{(a,b)}$ | $\mathrm{CS}[\rho_{\alpha=0}] - \mathrm{CS}[\rho]$ |
|---|---|---|---|---|---|
| $(0,0)$ | $(\frac{1}{2}, \frac{1}{2}, \frac{1}{2}, \frac{1}{2})$ | $\frac{53}{120}$ | $\frac{1}{4}(5 - \sqrt{5})$ | $\mathcal{O}_{(4,1)} = \mathcal{O}_{(1,1)}$ | $0$ |
| $(2,0)$ | $(\frac{3}{2}, \frac{1}{2}, \frac{1}{2}, \frac{1}{2})$ | $\frac{77}{120}$ | $\frac{1}{4}(5 + \sqrt{5})$ | $\mathcal{O}_{(2,1)} = \mathcal{O}_{(3,1)}$ | $\frac{4}{5}$ |

**Table 4**. Lee-Yang CFT from $S^2((5,3), (2,1), (3,1))$. The result is compatible with that $S_{(1,1),(1,1)} = \sqrt{\frac{2}{5-\sqrt{5}}}, S_{(1,1),(3,1)} = \sqrt{\frac{2}{5+\sqrt{5}}}$ and $h_{(3,1)} = -\frac{1}{5}$.

**Example:** $M(5,4) = $ (**Tricritical Ising model**) from $S^2((5,-1), (4,5), (3,1))$ We choose $(R, S) = (6,5)$. There are 6 irreducible $SL(2, \mathbb{C})$ characters $\rho_{\vec{j}}$:

| $(j_1, j_2)$ | $(\vec{n}, \lambda)$ | $\mathrm{CS}[\rho]$ | $\mathrm{Tor}[\rho]$ | $\mathcal{O}_{(a,b)}$ | $\mathrm{CS}[\rho_{\alpha=0}] - \mathrm{CS}[\rho_\alpha]$ |
|---|---|---|---|---|---|
| $(0,0)$ | $\left(\frac{1}{2}, \frac{1}{2}, \frac{1}{2}, \frac{1}{2}\right)$ | $\frac{37}{240}$ | $\frac{5}{2\left(\frac{5}{8} - \frac{\sqrt{5}}{8}\right)}$ | $\mathcal{O}_{(4,1)} = \mathcal{O}_{(1,3)}$ | $\frac{1}{2}$ |
| $(2,0)$ | $\left(\frac{3}{2}, \frac{1}{2}, \frac{1}{2}, \frac{1}{2}\right)$ | $\frac{133}{240}$ | $\frac{5}{2\left(\frac{5}{8} + \frac{\sqrt{5}}{8}\right)}$ | $\mathcal{O}_{(2,1)} = \mathcal{O}_{(3,3)}$ | $\frac{1}{10}$ |
| $(0,2)$ | $\left(\frac{1}{2}, \frac{3}{2}, \frac{1}{2}, \frac{1}{2}\right)$ | $\frac{157}{240}$ | $\frac{5}{2\left(\frac{5}{8} - \frac{\sqrt{5}}{8}\right)}$ | $\mathcal{O}_{(4,3)} = \mathcal{O}_{(1,1)}$ | $0$ |
| $(2,2)$ | $\left(\frac{3}{2}, \frac{3}{2}, \frac{1}{2}, \frac{1}{2}\right)$ | $\frac{13}{240}$ | $\frac{5}{2\left(\frac{5}{8} + \frac{\sqrt{5}}{8}\right)}$ | $\mathcal{O}_{(2,3)} = \mathcal{O}_{(3,1)}$ | $\frac{3}{5}$ |
| $(1,1)$ | $(1, 1, 1, 0)$ | $\frac{37}{60}$ | $\frac{5}{4\left(\frac{5}{8} + \frac{\sqrt{5}}{8}\right)}$ | $\mathcal{O}_{(2,2)} = \mathcal{O}_{(3,2)}$ | $\frac{3}{80}$ |
| $(3,1)$ | $(2, 1, 1, 0)$ | $\frac{13}{60}$ | $\frac{5}{4\left(\frac{5}{8} - \frac{\sqrt{5}}{8}\right)}$ | $\mathcal{O}_{(4,2)} = \mathcal{O}_{(1,2)}$ | $\frac{7}{16}$ |

**Table 5**. Tricritical Ising model from $S^2((5,-1), (4,5), (3,1))$. The result is compatible with the modular data of the tricritical Ising model, which can be evaluated from (2.4) and (2.7)

## 3 Field theory description of $\mathcal{T}_{(P,Q)}$

We now present a concrete and unified field-theoretic description for the $T_{\mathrm{irred}}[S^2(\vec{p}, \vec{q})]$ theory. By specializing the values of $((p_1, q_1), (p_2, q_2), (p_3, q_3))$ to $((P, P-R), (Q, S), (3, 1))$, the theory becomes the $\mathcal{T}_{(P,Q)}$ theory as proposed in (2.24). See (3.22) for the proposed $\mathcal{T}_{(P,Q)}$.

In principle, the theory can be constructed using the general algorithm proposed in [24, 25], which is based on a Dehn surgery representation of the 3-manifold using a hyperbolic link and an ideal triangulation of the link complement. As seen from various examples explored in [41], however, one needs to consider different hyperbolic links for each $T_{\mathrm{irred}}[S^2(\vec{p}, \vec{q})]$. This makes it difficult to succinctly describe the field theories for all $S^2(\vec{p}, \vec{q})$ in a unified manner.

### 3.1 Field theory description of $T_{\mathrm{irred}}[S^2(\vec{p}, \vec{q})]$

The Seifert fiber space $S^2(\vec{p}, \vec{q})$ depicted in Figure 1 can be alternatively represented as follows:

$$S^2(\vec{p}, \vec{q}) = \left((\Sigma_{0,3} \times S^1) \bigcup_{i=1}^3 (D_2 \times S^1)_i\right) / \sim \tag{3.1}$$

$$\text{with } p_i A_i + q_i B_i \sim [S^1] \subset H_1[\partial D_2, \mathbb{Z}], \quad i = 1, 2, 3.$$

Here, $\Sigma_{g=0,h=3}$ denotes a three-punctured sphere and $(A_i, B_i)$ are basis elements of $H_1(\partial(\Sigma_{0,3} \times S^1), \mathbb{Z})$m where $A_i$ (resp. $B_i$) represents the 1-cycle circling the $i$-th puncture (resp. the 1-cycle along the $S^1 \in \partial(\Sigma_{0,3}) \times S^1$). The equivalence relation $p_i A_i + q_i B_i \simeq [S^1] \in H_1[\partial D_2, \mathbb{Z}]$ corresponds to the relation $x_i^{p_i} h^{q_i} = 1$ in the fundamental group (2.29) and the relation $x_1 x_2 x_3 = 1$ comes from the same relation in $\pi_1(\Sigma_{0,3})$.

Using the geometrical representation, the field theory $T[S^2(\vec{p}, \vec{q})]$ can be constructed as follows, see Figure 2. First, we prepare the $T[\Sigma_{0,3} \times S^1]$ theory which is expected to a $\mathcal{N} = 4$ theory with three $SU(2)$ flavor symmetries associated with the 3 boundary

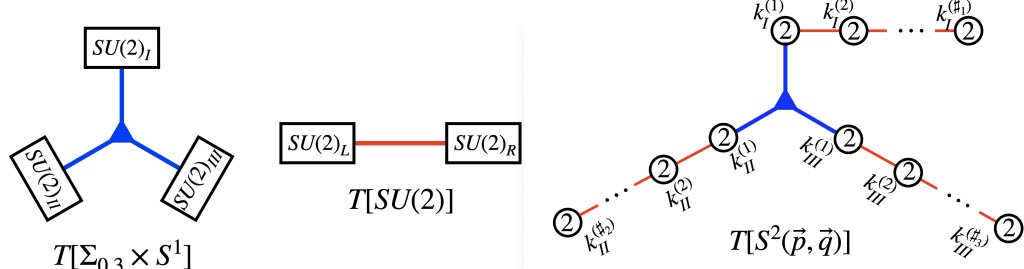

**Figure 2**. Generalized quiver diagrams for $T[\Sigma_{0,3} \times S^1]$, $T[SU(2)]$ and $T[S^2(\vec{p}, \vec{q})]$. The difference between $T_{\text{full}}[S^2(\vec{p}, \vec{q})]$ and $T_{\text{irred}}[S^2(\vec{p}, \vec{q})]$ arises only from different choices of the $T[\Sigma_{0,3} \times S^1]$ theory, either $T_{\text{full}}[\Sigma_{0,3} \times S^1]$ or $T_{\text{irred}}[\Sigma_{0,3} \times S^1]$.

tori (punctures$\times S^1$). The theory is depicted by a blue trivalent vertex with 3 legs, and the boxes attached to the legs represent the three $SU(2)$ flavor symmetries. Gluing the solid torus is a Dehn filling procedure, whose corresponding field-theoretic operation in 3D-3D correspondence has been studied in literature [25, 36, 39, 48, 49]. In the Dehn filling operation, the $T[SU(2)]$ theory [50] plays an important role. The $T[SU(2)]$ is a $\mathcal{N} = 4$ theory with $SU(2)_L \times SU(2)_R$ flavor symmetry, (see Appendix A for details) and is depicted by a red line with two boxes representing the two $SU(2)$ symmetries. The Dehn filling operation, $p_i A_i + q_i B_i \sim [S_1] \in H_1[\partial D_2, \mathbb{Z}]$, at each $i$ corresponds to coupling the $T[\Sigma_{0,3} \times S^1]$ theory to $(\sharp - 1)$-copies of $T[SU(2)]$ theories using the $i$-th $SU(2)$ flavor symmetry in $T[\Sigma_{0,3} \times S^1]$ and $2(\sharp - 1)$ $SU(2)$s in the $T[SU(2)]^{\otimes(\sharp-1)}$, as described in 3rd quiver diagram in Figure 2. The circle denotes the $\mathcal{N} = 3$ gauging the diagonal $SU(2)$, and the integer $k$ next to the circle denotes the Chern-Simons (CS) level. The number of $T[SU(2)]$s, $\sharp - 1$, and the CS levels $\vec{k} = (k^{(1)}, \ldots, k^{(\sharp)})$ are related to the Dehn filling slope $(p, q)$ as follows:

$$\frac{q}{p} = \cfrac{1}{k^{(1)} - \cfrac{1}{k^{(2)} - \cfrac{1}{k^{(3)} - \cdots \frac{1}{k^{(\sharp)}}}}} \ .$$

or equivalently,

$$g(\vec{k}) := T^{k^{(1)}} S T^{k^{(2)}} \ldots S T^{k^{(\sharp)}} = \pm \begin{pmatrix} q & * \\ p & * \end{pmatrix} \ .$$

(3.2)

Here $S$ and $T \in SL(2, \mathbb{Z})$ are chosen as

$$S = \begin{pmatrix} 0 & -1 \\ 1 & 0 \end{pmatrix}, \quad T = \begin{pmatrix} 1 & 0 \\ 1 & 1 \end{pmatrix} \ .$$

(3.3)

Especially when $p/q$ is an integer, i.e., $q = 1$, the Dehn filling operation corresponds to the gauging of $SU(2)$ flavor symmetry with CS level $p$.

**Field theory for** $T_{\mathrm{irred}}\left[S^2((p_1,q_1),(p_2,q_2),(p_3,q_3))\right]$ Using the prescription above, the field theory $T_{\mathrm{full}}[S^2(\vec{p},\vec{q})]$ is constructed as follows [37, 39, 51]

$$
\begin{aligned}
T_{\mathrm{full}}[\Sigma_{0,3} \times S^1] &= (S^1\text{-reduction of 4D } T_{N=2} \text{ theory}) \\
&= (\text{a free theory with 8 half-hypermultiplets in } \mathbf{2}\otimes\mathbf{2}\otimes\mathbf{2} \text{ under the } SU(2)^3) \,.
\end{aligned}
\tag{3.4}
$$

The $T_{\mathrm{irred}}[S^2(\vec{p},\vec{q})]$ can be constructed in the same way except that the $T_{\mathrm{full}}[\Sigma_{0,3} \times S^1]$ should be replaced by $T_{\mathrm{irred}}[\Sigma_{0,3} \times S^1]$. We propose that:

$$
T_{\mathrm{irred}}\left[\Sigma_{0,3} \times S^1\right] = (\text{a topological field theory}) \,,
\tag{3.5}
$$

This proposal is based on the observation that $T_{\mathrm{irred}}[M]$ theories for non-hyperbolic 3-manifolds $M$ with torus boundaries usually exhibit a mass gap and flow to topological field theory. As we will see below, this proposal passes several non-trivial consistency checks. Combining the above proposal with the general Dehn surgery prescription in 3D-3D correspondence, we propose that

$$
\begin{aligned}
&T_{\mathrm{irred}}\left[M = S^2((p_1,q_1),(p_2,q_2),(p_3,q_3))\right] \otimes (\text{a unitary TQFT}) \\
&\simeq \left[(\mathcal{D}(p_1,q_1) \otimes \mathcal{D}(p_2,q_2) \otimes \mathcal{D}(p_3,q_3))\right]/H^1(M,\mathbb{Z}_2) \,.
\end{aligned}
\tag{3.6}
$$

Here $\sim$ denotes the stronger IR equivalence in (2.27). Here $H^1(M,\mathbb{Z}_2)$ represents the 1-form symmetry in $\prod_i \mathcal{D}(p_i,q_i)$ which geometrically originates from the $\mathbb{Z}_2$ cohomology of the internal 3-manifold in 3D-3D correspondence [37, 45]. The theory $\mathcal{D}(p,q)$ is defined as follows:

$$
D(\vec{k}) \simeq \mathcal{D}(p,q) \otimes \mathrm{TFT}[\vec{k}], \quad \text{where}
$$

$$
D(\vec{k}) := \begin{cases} \dfrac{T[SU(2)]^{\otimes(\sharp-1)}}{SU(2)^{(1)}_{k^{(1)}} \otimes SU(2)^{(2)}_{k^{(2)}} \cdots \times SU(2)^{(\sharp)}_{k^{(\sharp)}}}, & \sharp \geq 2 \\ \mathcal{N}=2 \ \text{pure } SU(2)_{k^{(1)}} \text{ CS theory}, & \sharp = 1 \end{cases}
\tag{3.7}
$$

Here $/G_k$ denotes $\mathcal{N}=3$ gauging of $G$ symmetry with Chern-Simon level $k$. The CS levels $\vec{k} = (k^{(1)}, \ldots, k^{(\sharp)})$ are related to the $(p,q)$ as in (3.2). The gauged $SU(2)$ symmetries are

$$
\begin{aligned}
&SU(2)^{(1)} \ : \ SU(2)^{(1)}_L := SU(2)_L \text{ of the 1st } T[SU(2)] \,, \\
&SU(2)^{(2 \leq I \leq \sharp-1)} \ : \ \text{diagonal subgroup of } (SU(2)^{(I-1)}_R \times SU(2)^{(I)}_L) \,, \\
&SU(2)^{(\sharp)} \ : \ SU(2)^{(\sharp-1)}_R \,.
\end{aligned}
\tag{3.8}
$$

The field theory description can be summarized in the quiver diagram in Figure 3. The $D(\vec{k})$

**Figure 3**. A quiver diagram for $D(\vec{k})$ theory.

theory has $(\mathbb{Z}_2)^{\otimes\sharp}$ 1-form symmetry originating from the center subgroup of $SU(2)^\sharp$ gauge

symmetry. The symmetry has non-trivial 't Hooft anomalies [25] which can be characterized by the following action of the 4D anomaly theory

$$S_{\text{anoamly}} = \pi \int_{\mathcal{M}_4} \left( \sum_{I=1}^{\sharp} k^{(I)} \frac{\mathcal{P}(\omega_2^{(I)})}{2} + \sum_{J=1}^{\sharp-1} \omega_2^{(J)} \cup \omega_2^{(J+1)} \right) . \tag{3.9}$$

Here $\omega_2^{(I)} \in H^2(\mathcal{M}_4, \mathbb{Z}_2)$ is the two-form background field for each $\mathbb{Z}_2$ 1-form symmetry from $I$-th $SU(2)$ gauge symmetry and $\mathcal{P}$ is the Pontryagin square operation. By matching the anomaly with that of the decoupled topological field theory [52], we expect that the decoupled topological theory is given in the the following form:

$\text{TFT}[\vec{k} = (k^{(1)}, k^{(2)}, \dots, k^{(\sharp)})]$

$$= U(1)_{\mathcal{K}}^{\sharp} \text{ theory with mixed CS level } \mathcal{K}_{IJ} = 2 \times \begin{cases} +1 \text{ or } -1, & |I - J| = 1 \\ 0, & I = J \text{ and } k^{(I)} \in 2\mathbb{Z} \\ +1 \text{ or } -1, & I = J \text{ and } k^{(I)} \in 2\mathbb{Z} + 1 \\ 0, & |I - J| > 0 \end{cases}.$$

$$\tag{3.10}$$

Refer to [53] for a similar analysis done for $T_{\text{full}}[S^2((k_1, 1), (k_2, 1), (k_3, 1))]$ theory. See also Appendix B for more details of the decoupled TQFT. By removing the decoupled TQFT from $D(\vec{k})$, we obtain the $\mathcal{D}(p, q)$. We expect following properties of the $\mathcal{D}(p, q)$ theory

$i)$ $\mathcal{D}(p, q)$ does not depend on the choice of $\vec{k}$ for given $(p, q)$. ,

$ii)$ $\mathcal{D}(p, q) \simeq \mathcal{D}(p, q + p\mathbb{Z})$ . $\tag{3.11}$

The first property follows from the basic IR dualities (modulo topological sectors) depicted

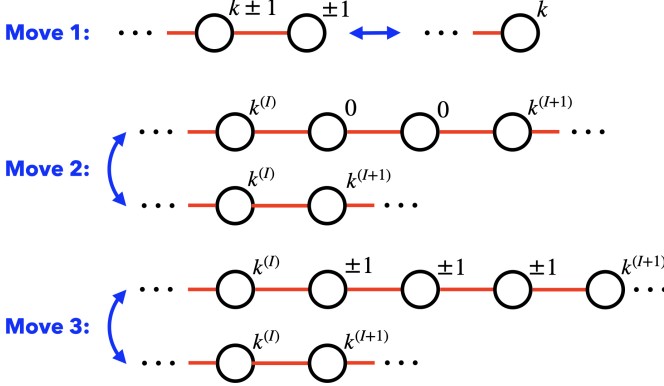

**Figure 4**. Three basic dualities (modulo topological sectors) guaranteeing the independence of $D(p, q)$ on the choices of $\vec{k}$. The 2nd and 3rd moves follow from the $SL(2, \mathbb{Z})$ structure of the duality wall theory. The 1st move comes from the IR duality between $(T[SU(2)]/(SU(2)_R)_{\pm 1}) \sim$ (empty theory only with background CS level $\mp 1$ for $SU(2)_L$). See the computation in (A.19) for a non-trivial check of the duality using superconformal index. The same moves were used in constructing $T_{\text{full}}[M]$ for 3-manifolds $M$ associated to plumbed graphs in [36].

in Figure 4, which imply that

$$D(\vec{k}_1) \sim D(\vec{k}_2) \quad \text{if } g(\vec{k}_1) = g(\vec{k}_2) \cdot T^{\pm 1} S T^{\pm 1} . \tag{3.12}$$

Note that $g(\vec{k}) = \pm \begin{pmatrix} q & * \\ p & * \end{pmatrix}$ and the right multiplication of $T^{\pm 1} S T^{\pm 1} = \begin{pmatrix} \mp 1 & -1 \\ 0 & \mp 1 \end{pmatrix}$ does not change the slope $(p,q)$. The 2nd property follows from the move 1 in the figure with reversed left/right orientation on each quivers, which implies that

$$D(\vec{k}_1) \sim D(\vec{k}_2) \quad \text{if } g(\vec{k}_1) = T^{\pm 1} S T^{\pm 1} \cdot g(\vec{k}_2) . \tag{3.13}$$

The left multiplication of $T^{\pm 1} S T^{\pm 1}$ changes the $(p,q)$ to $(p, q \pm p)$. The above IR equivalence $\sim$ modulo topological sector can promoted into the IR equivalence $\simeq$ in (2.27) if the decoupled TFT$(\vec{k})$s on both sides are removed.

From the two properties in (3.11), it is easy to see that (for $|p| \geq 2$):

> If $q = \pm 1 \pmod{p}$,
>
> the $\mathcal{D}(p,q)$ has a mass gap and flows to a unitary TQFT in the IR
> $\tag{3.14}$

since

$$\mathcal{D}(p, \pm 1 + p\mathbb{Z}) \simeq \mathcal{D}(p, \pm 1) \sim D(\vec{k} = (\pm p)) = (\mathcal{N} = 3 \ SU(2)_{\pm p})$$
$$\simeq (\mathcal{N} = 2 \ SU(2)_{\pm p}) \simeq SU(2)_{\pm p - 2 \times \text{sign}(\pm p)} .$$

The pure $\mathcal{N} = 3$ CS theory with non-zero CS level is IR equivalent to pure $\mathcal{N} = 2$ CS theory since the adjoint chiral mutiplet in the $\mathcal{N} = 3$ mutiplet has a superpotential mass term and can be integrated out. The pure $\mathcal{N} = 2$ Chern-Simons theory $SU(2)_k$ contains an auxiliary massive gaugino, and integrating out it induces a CS level shift by $-2 \times \text{sign}(k)$. Furthermore, one can check that

$$\mathcal{D}(2, 1 + 2\mathbb{Z}) \simeq \mathcal{D}(3, 1 + 3\mathbb{Z}) \simeq (\text{a trivial theory}), \tag{3.15}$$

The first follows from the fact that $SU(2)_{2-2} = SU(2)_0$ is a trivial theory, while the 2nd follows from that $SU(2)_{3-2} = SU(2)_1 \simeq U(1)_2$ and TFT$[\vec{k} = (3)]$ in (3.10) is again $U(1)_2$. One the other hand, we expec that

> If $q \neq \pm 1 \pmod{p}$,
>
> the $\mathcal{D}(p,q)$ flows to an $\mathcal{N} = 4$ rank-0 SCFT in the IR.
> $\tag{3.16}$

The non-zero CS terms lift all the Coulomb/Higgs branches of $T[SU(2)]$ theories. The UV $\mathcal{N} = 3$ will be enhanced to $\mathcal{N} = 4$ in the IR thanks to the nilpotency properties of moment maps of the $T[SU(2)]$ theory [12, 50, 54]. Along with the proposal in (3.6), (3.14) and (3.16) are compatible with the expected IR phases of $T[S^2(\vec{p}, \vec{q})]$ given in (2.18).

**Comparison with $T_{\mathrm{irred}}[S^2((\vec{p},\vec{q}))]$s in [41]**   Based on a Dehn surgery representation of $S^2((\vec{p},\vec{q}))$ with hyperbolic knots, the field theories of $T_{\mathrm{irred}}[S^2((\vec{p},\vec{q}))]$ for various $(\vec{p},\vec{q})$s were analyzed in [41]. For instance, the 3D theory $T_{\mathrm{irred}}[M]$ for $M = (S^3\backslash\mathbf{5}_2)_p$ (the 3-manifold obtained by a Dehn surgery on $\mathbf{5}_2$ knot, a.k.a the 3-twist knot, with an integral slop $p$) is given by

$$
T_{\mathrm{irred}}[(S^3\backslash\mathbf{5}_2)_p]
$$

$$
=\begin{cases}
\dfrac{\left(U(1)_{-\frac{1}{2}}\times SU(2)_{p-2}\ \mathcal{N}=2 \text{ gauge theory coupled to a chiral in } (\mathrm{Adj})_{+1}\right)}{\mathbb{Z}_2}\,, & p\in 2\mathbb{Z} \\[2em]
\dfrac{(U(1)_{-\frac{1}{2}}\times SU(2)_{p-2}\ \mathcal{N}=2 \text{ gauge theory coupled to a chiral in } (\mathrm{Adj})_{+1})\otimes U(1)_{\pm 2}}{\mathbb{Z}_2^{\mathrm{diag}}}\,, & p\in 2\mathbb{Z}+1
\end{cases}
\tag{3.17}
$$

The theory in the numerator has a $\mathbb{Z}_2$ 1-form symmetry originating from the center $\mathbb{Z}_2$ subgroup of the $SU(2)$ gauge group. For even $p$, the $\mathbb{Z}_2$ 1-form symmetry is non-anomalous (i.e., having a trivial 't Hooft anomaly) and thus can be gauged. For odd $p$, however, the 1-form symmetry has a non-trivial 't Hooft anomaly, requiring it to be tensored with the $U(1)_{\pm 2}$ theory, which has the same anomalous 1-form symmetry, before gauging the non-anomalous diagonal $\mathbb{Z}_2$ 1-form symmetry.

Topologically it is known that [55]

$$
(S^3\backslash\mathbf{5}_2)_p = \begin{cases}
S^2((2,1),(3,1),(11,-9))\,, & p=1 \\
S^2((2,1),(4,1),(7,-5))\,, & p=2 \\
S^2((3,1),(3,1),(5,-3))\,, & p=3
\end{cases}
\tag{3.18}
$$

According to our proposal in (3.6), we expect the following dual description[3]

$$
T_{\mathrm{irred}}[(S^3\backslash\mathbf{5}_2)_p] \sim \begin{cases}
D(\vec{k}=(-1,4,-2))\,, & p=1 \\
D(\vec{k}=(-1,2,-2))\,, & p=2 \\
D(\vec{k}=(-2,-3))\,, & p=3
\end{cases}
\tag{3.19}
$$

Note that $\mathcal{D}(p,1)$ with $p\in\mathbb{Z}$ is a (unitary) TQFT and thus ignored in the above. Using the expression in (A.17), one can compute the superconformal indices:

$$
\mathcal{I}^{\mathrm{sci}}_{D(\vec{k})}(\eta,\nu=0)\quad\text{with }\vec{k}=(-1,4,-2)
$$
$$
=1-q-\left(\eta+\frac{1}{\eta}\right)q^{3/2}-2q^2-\eta^{-1}q^{5/2}+(\eta^2-1)q^3+\left(\eta-\frac{1}{\eta}\right)q^{7/2}+0q^4+\dots
$$
$$
\mathcal{I}^{\mathrm{sci}}_{D(\vec{k})}(\eta,\nu=0)\quad\text{with }\vec{k}=(-1,2,-2)
$$
$$
=1-q-\left(\eta+\frac{1}{\eta}\right)q^{3/2}-2q^2-\eta^{-1}q^{5/2}+(\eta^2-1)q^3+\left(\eta-\frac{1}{\eta}\right)q^{7/2}+\eta^2 q^4+\dots
$$
$$
\mathcal{I}^{\mathrm{sci}}_{D(\vec{k})}(\eta,\nu=0)\quad\text{with }\vec{k}=(-2,-3)
$$
$$
=1-q-\left(\eta+\frac{1}{\eta}\right)q^{3/2}-2q^2-\left(\eta+\frac{1}{\eta}\right)q^{5/2}-2q^3-\left(\eta+\frac{1}{\eta}\right)q^{7/2}-2q^4+\dots\ .
\tag{3.20}
$$

---

[3]Using $\frac{-9}{11}=\frac{1}{-1-\frac{1}{4-\frac{1}{-2}}}$, $\frac{-5}{7}=\frac{1}{-1-\frac{1}{2-\frac{1}{-2}}}$ and $-\frac{3}{5}=\frac{1}{-2-\frac{1}{-3}}$.

These results match the superconformal indices given in [41] for the $T_{\text{irred}}[(S^3 \backslash \mathbf{5}_2)_p]$ theory with $p = 1, 2, 3$ respectively. For rank-0 SCFTs, the superconformal index is defined as:

$$\mathcal{I}^{\text{sci}}(\eta, \nu) := \text{Tr}(-1)^{R_\nu} q^{\frac{R_\nu}{2} + j_3} \eta^A , \qquad (3.21)$$

where $A$ and $R_\nu$ are defined in (2.13) and (2.14).

## 3.2 Field theory for $\mathcal{T}_{(P,Q)}$

Combining equations (2.24) and (3.6), we arrive at the final form of the field theory description:

$$\mathcal{T}_{(P,Q)} \simeq T_{\text{irred}}[S^2((P, P - R), (Q, S), (3, 1))]$$
$$\simeq \begin{cases} (\mathcal{D}(P, P - R) \otimes \mathcal{D}(Q, S) \otimes U(1)_{\pm 2}, & P \text{ and } Q \in 2\mathbb{Z} + 1 \\ (\mathcal{D}(P, P - R) \otimes \mathcal{D}(Q, S) \otimes U(1)_2 \times U(1)_{\pm 2}) / \mathbb{Z}_2^{\text{diag}}, & \text{otherwise} \end{cases} \qquad (3.22)$$

The theory $\mathcal{D}(p, q)$ is defined in (3.7). **This is the main result of our paper**. In the first case, we include the decoupled topological field theory, $U(1)_2$ or $U(1)_{-2}$, in order that (recall that $\mathcal{Z}_b^{\text{con}} := \mathcal{Z}^{S_b^3}(M = 0, \nu = 0)$)

$$| \left( \mathcal{Z}_{b=1}^{\text{con}} \text{ of } \mathcal{T}_{(P,Q)} \right) | = | \left( \mathcal{Z}_{b=1}^{\text{con}} \text{ of } \mathcal{D}(P, P - R) \right) \times \left( \mathcal{Z}_{b=1}^{\text{con}} \text{ of } \mathcal{D}(Q, S) \right) \times \frac{1}{\sqrt{2}} |$$
$$= \sqrt{\frac{8}{PQ}} \sin \left( \frac{\pi}{P} \right) \sin \left( \frac{\pi}{Q} \right) = (\min_\alpha |S_{0\alpha}| \text{ of } M(P, Q)) . \qquad (3.23)$$

We use the result in (B.18) and the fact that the $S^3$ partition function of $U(1)_\pm$ is $\frac{1}{\sqrt{2}}$. The decoupled $U(1)_{\pm 2}$ has a $\mathbb{Z}_2$ 1-form symmetry generated by an anyon, a topological line defect, with topological spin $\pm \frac{1}{4}$ (mod 1), which is compatible with the fact that there is a primary operator with conformal dimension $\pm \frac{1}{4}$ (mod 1) in the $M(P, Q)$ with $P, Q \in 2\mathbb{Z} + 1$.

When $P$ (resp. $Q$) is even, the theory $\mathcal{D}(P, P - R)$ (resp. $\mathcal{D}(Q, S)$) has an non-anomalous $\mathbb{Z}_2$ 1-form symmetry generated by an anyon with topological spin 0 or 1/2 [52], see (B.19). The theory in the numerator has three $\mathbb{Z}_2$ 1-form symmetries and $\mathbb{Z}_2^{\text{diag}}$ denotes the gauging the diagonal one. The $\pm$ in the $U(1)_2 \times U(1)_{\pm 2}$ is chosen such that the $\mathbb{Z}_2^{\text{diag}}$ 1-form symmetry is bosonic (i.e., the symmetry generating anyon has a topological spin 0).

When $Q = P + 1$, the $(R, S)$ can be chosen as $(-1, -1)$ and the $\mathcal{T}_{(P,Q)}$ theory becomes

$$(\mathcal{D}(P, P + 1) \otimes \mathcal{D}(P + 1, -1) \otimes U(1)_2 \otimes U(1)_{\pm 2}) / \mathbb{Z}_2^{\text{diag}} ,$$
$$\simeq \frac{SU(2)_{(P-2)} \otimes SU(2)_{-(P-1)} \otimes U(1)_2}{\mathbb{Z}_2^{\text{diag}}} . \qquad (3.24)$$

This is the coset description of unitary minimal model $\mathcal{T}_{(P,P+1)}$. We use the fact that

For $P \in 2\mathbb{Z}_{\geq 1}$, $\mathcal{D}(P + 1, -1) \otimes U(1)_{\pm 2} \simeq SU(2)_{-(P-1)}$ and $\mathcal{D}(P, P + 1) \simeq SU(2)_{P-2}$ ,

For $P \in 2\mathbb{Z}_{\geq 1} + 1$, $\mathcal{D}(P + 1, -1) \simeq SU(2)_{-(P-1)}$ and $\mathcal{D}(P, P + 1) \otimes U(1)_{\mp 2} \simeq SU(2)_{(P-2)}$ .

The 2D chiral central charge of the theory can be computed as follows using that $c_{2d}(SU(2)_k) = \frac{3k}{|k|+2}$ and $c_{2d}(U(1)_2) = 1$,

$$c_{2d} = \frac{3(P-2)}{P} - \frac{3(P-1)}{P+1} + 1 = 1 - \frac{6P}{P(P+1)} , \qquad (3.25)$$

which matches (2.2).

**Example :** $(P,Q) = (3,5)$ Choosing $(R,S) = (1,2)$ and using equations (3.22) and (3.15), we find:

$$\mathcal{T}_{(P,Q)} \simeq \mathcal{D}(3,2) \otimes \mathcal{D}(5,2) \otimes U(1)_{\pm 2} \simeq \mathcal{D}(5,2) \otimes U(1)_{\pm 2} . \qquad (3.26)$$

From that $\frac{2}{5} = \frac{1}{3-\frac{1}{2}}$, we obtain

$$\mathcal{D}(5,2) \otimes \mathrm{TFT}[\vec{k}] \simeq D(\vec{k}) \text{ with } \vec{k} = (3,2) . \qquad (3.27)$$

Utilizing the explicit computation from (A.24), we confirm that the set $\mathbf{HF} := (\mathcal{H}^{-1/2}, \mathcal{F}/\mathcal{F}_{\alpha=0})$ of the $D((3,2))$ theory in the $A$-twisting limit can be factorized as follows:

$\mathbf{HF}^A$ of $D((3,2))$

$$= \left\{ \left( \frac{4}{\sqrt{15}} \sin(\frac{\pi}{3}) \sin(\frac{2\pi}{5}), 1 \right), \left( \frac{4}{\sqrt{15}} \sin(\frac{\pi}{3}) \sin(\frac{\pi}{5}), e^{\frac{2\pi i}{5}} \right) \right\} \times \left\{ \left( \frac{1}{2}, 1 \right)^{\otimes 2}, \left( \frac{1}{2}, i \right), \left( \frac{1}{2}, \frac{1}{i} \right) \right\} . \qquad (3.28)$$

The second factor can be interpreted as the contribution from the decoupled $\mathrm{TFT}[\vec{k}]$. By producing the first factor, which is from $\mathcal{D}(5,2)$, with the $\mathbf{HF} = \{(|S_{0\alpha}|, e^{2\pi i h_\alpha})\}$ of $U(1)_{-2}$, we obtain the set for the $\mathcal{T}_{(3,5)}$ theory:

$\mathbf{HF}^A$ of $\mathcal{T}_{(3,5)}$

$$= \left\{ \left( \frac{4}{\sqrt{15}} \sin(\frac{\pi}{3}) \sin(\frac{2\pi}{5}), 1 \right), \left( \frac{4}{\sqrt{15}} \sin(\frac{\pi}{3}) \sin(\frac{\pi}{5}), e^{\frac{2\pi i}{5}} \right) \right\} \times \left\{ \left( \frac{1}{\sqrt{2}}, 1 \right), \left( \frac{1}{\sqrt{2}}, \frac{1}{i} \right) \right\} . \qquad (3.29)$$

This set nicely matches with the set of $(|S_{0\alpha}|, e^{2\pi i h_\alpha})$ for $M(3,5)$, as expected from the dictionaries in Table 1.

Alternatively, using $\frac{2}{5} = \frac{1}{2-\frac{1}{-2}}$, we have:

$$\mathcal{D}(5,-2) \otimes \mathrm{TFT}[\vec{k}] \simeq D(\vec{k}) \text{ with } \vec{k} = (2,-2) . \qquad (3.30)$$

In this case, we find:

$\mathbf{HF}^A$ of $D((2,-2))$

$$= \left\{ \left( \frac{4}{\sqrt{15}} \sin(\frac{\pi}{3}) \sin(\frac{2\pi}{5}), 1 \right), \left( \frac{4}{\sqrt{15}} \sin(\frac{\pi}{3}) \sin(\frac{\pi}{5}), e^{\frac{2\pi i}{5}} \right) \right\} \times \left\{ \left( \frac{1}{2}, 1 \right)^{\otimes 3}, \left( \frac{1}{2}, -1 \right) \right\} . \qquad (3.31)$$

Again, the second factor can be interpreted as the contribution from the decoupled $\mathrm{TFT}[\vec{k}]$. By omitting the topological factor, we obtain the set for $\mathcal{D}(5,2)$. It is equivalent to the previous set obtained using $D((3,2))$.

**Example :** $(P, Q) = (3, 8)$   Choosing $(R, S) = (1, 3)$, we have:

$$\mathcal{T}_{(3,8)} \simeq \left(\mathcal{D}(3, 2) \otimes \mathcal{D}(8, 3) \otimes U(1)_2 \times U(1)_{\pm 2}\right) / \mathbb{Z}_2^{\text{diag}} \simeq \left(\mathcal{D}(8, 3) \otimes U(1)_2 \times U(1)_{\pm 2}\right) / \mathbb{Z}_2^{\text{diag}} \, . \tag{3.32}$$

Using $\frac{3}{8} = \frac{1}{3 - \frac{1}{3}}$, we obtain:

$$\mathcal{D}(8, 3) \otimes \text{TFT}[\vec{k}] \simeq D(\vec{k}) \text{ with } \vec{k} = (3, 3) \, . \tag{3.33}$$

Using the explicit computation in (A.24), we confirm that the set $\mathbf{HF} := \{(\mathcal{H}^{-1/2}, \mathcal{F}/\mathcal{F}_{\alpha=0})\}$ of $D((3, 3))$ theory in the $A$-twisting limit can be factorized as follows:

$\mathbf{HF}^A$ of $D((3, 3))$

$$= \left\{ \left(\frac{1}{2}, e^{\frac{29i\pi}{16} - \frac{\pi i}{2}}\right), \left(\frac{1}{2\sqrt{2}}, e^{\frac{25i\pi}{16} - \frac{\pi i}{2}}\right), \left(\frac{1}{2\sqrt{2}}, e^{\frac{25i\pi}{16} - \frac{\pi i}{2}}\right), \left(\frac{1}{2} \sin\left(\frac{\pi}{8}\right), i\right), \right.$$
$$\left. \left(\frac{1}{2} \sin\left(\frac{\pi}{8}\right), \frac{1}{i}\right), \left(\frac{1}{2} \cos\left(\frac{\pi}{8}\right), 1\right), \left(\frac{1}{2} \cos\left(\frac{\pi}{8}\right), -1\right) \right\} \times \left\{ \left(\frac{1}{\sqrt{2}}, 1\right), \left(\frac{1}{\sqrt{2}}, i\right) \right\} \, . \tag{3.34}$$

The 2nd factor can be regarded as the contribution from the decoupled $\text{TFT}[\vec{k}]$ and 1st factor is from $\mathcal{D}(8, -3)$. From the $\mathbf{HF}^A$ of $\mathcal{D}(8, -3)$, one can see that the $\mathcal{D}(8, -3)$ theory has a non-anomalous $\mathbb{Z}_2$ 1-form symmetry generated by an anyon with topological spin $\frac{1}{2}$. Hence, we have the set for $\mathcal{T}_{(3,8)}$ theory:[4]

$$\mathbf{HF}^A \text{ of } \mathcal{T}_{(3,8)} = \left(\mathcal{D}(8, 3) \otimes U(1)_2 \otimes U(1)_2\right) / \mathbb{Z}_2^{\text{diag}}$$
$$= \left\{ \left(\frac{1}{2}, e^{\frac{29i\pi}{16}}\right), \left(\frac{1}{2\sqrt{2}}, e^{\frac{25i\pi}{16}}\right), \left(\frac{1}{2\sqrt{2}}, e^{\frac{25i\pi}{16}}\right), \left(\frac{1}{2} \sin\left(\frac{\pi}{8}\right), i\right), \right.$$
$$\left. \left(\frac{1}{2} \sin\left(\frac{\pi}{8}\right), \frac{1}{i}\right), \left(\frac{1}{2} \cos\left(\frac{\pi}{8}\right), 1\right), \left(\frac{1}{2} \cos\left(\frac{\pi}{8}\right), -1\right) \right\} \, . \tag{3.35}$$

This set nicely matches the set of $(|S_{0\alpha}|, e^{2\pi i h_\alpha})$ of $M(3, 8)$, as expected from the dictionaries in Table 1.

### 3.3   Comparison with the $\mathcal{T}_{(2,2t+3)}$ by Gang-Kim-Stubbs

For the case when $(P, Q) = (2, 2t + 3)$, the 3D $\mathcal{T}_{(P,Q)}$ is (we choose $R = 1, S = t + 2$)

$$\mathcal{T}_{(2,2t+3)} \simeq T_{\text{irred}}[S^2((2, 1), (2t + 3, t + 2), (3, 1))] \, ,$$
$$\simeq \mathcal{D}(2t + 3, t + 2) \otimes \frac{U(1)_2 \otimes U(1)_{-2}}{\mathbb{Z}_2^{\text{diag}}} \simeq \mathcal{D}(2t + 3, t + 2) \, . \tag{3.36}$$

---

[4]Let $\mathcal{T}$ be a theory with $\mathbb{Z}_2$ 1-form symmetry generated by an anyon with topological spin $1/2$. The Bethe vacua can be divided into two sets, $\mathcal{S}_{\text{BE}} = \mathcal{S}_{\text{BE}}^{\text{even}} \cup \mathcal{S}_{\text{BE}}^{\text{odd}}$. Two Bethe vacua in $\mathcal{S}_{\text{BE}}^{\text{even}}$, when related by the $\mathbb{Z}_2$ 1-form symmetry, have the same $\mathcal{H}$ but their $\mathcal{F}$ differ by a sign. On the other hand, two Bethe vacua (which can be identical) in $\mathcal{S}_{\text{BE}}^{\text{odd}}$, when related by the $\mathbb{Z}_2$ 1-form symmetry, have the same $\mathcal{H}$ and $\mathcal{F}$. In the theory $\tilde{\mathcal{T}} := \frac{\mathcal{T} \otimes U(1)_2 \otimes U(1)_2}{\mathbb{Z}_2^{\text{diag}}}$, the Bethe-vacua set can be canonically identified with that of $\mathcal{T}$. The $\mathcal{H}$ of Bethe-vacua in $\tilde{\mathcal{T}}$ are the same as in $\mathcal{T}$. For Bethe vacua in $\mathcal{S}_{\text{BE}}^{\text{even}}$, their $\mathcal{F}$s in $\tilde{\mathcal{T}}$ are equal to that in $\mathcal{T}$. However, for Bethe vacua in $\mathcal{S}_{\text{BE}}^{\text{odd}}$, the $\mathcal{F}$s in $\tilde{\mathcal{T}}$ differ by a factor of $\exp(\frac{\pi i}{2})$ compared to that in $\mathcal{T}$.

Here, we use the fact that both $\mathcal{D}(2,1)$ and $(U(1)_2 \otimes U(1)_{-2})/\mathbb{Z}_2^{\text{diag}}$ are trivial theories. Using $\frac{t+2}{2t+3} = \frac{1}{2 - \frac{1}{t+2}}$, the $\mathcal{D}(2t+3, t+2)$ is given as

$$D(\vec{k}) \simeq \mathcal{D}(2t+3, t+2) \otimes \text{TFT}[\vec{k}] \text{ with } \vec{k} = (2, t+2) . \tag{3.37}$$

The decoupled topological field theory is, see (B.9) and (B.11):

$$\text{TFT}[\vec{k} = (2, t+2)] \simeq U(1)_{\mathcal{K}}^2 \text{ with mixed CS level } \mathcal{K} = \begin{cases} \begin{pmatrix} 0 & 2 \\ 2 & 0 \end{pmatrix}, & t \in 2\mathbb{Z} \\ \begin{pmatrix} 2 & 2 \\ 2 & 0 \end{pmatrix}, & t \in 2\mathbb{Z} + 1 \end{cases} \tag{3.38}$$

Recently, an abelian $\mathcal{N} = 2$ gauge theory description, denoted $\mathcal{T}_{(2,2t+3)}^{GKS}$, of $\mathcal{T}_{(2,2t+3)}$ was proposed in [16]:

$$\mathcal{T}_{(2,2t+3)}^{GKS} = \left( \frac{(\mathcal{T}_\Delta)^{\otimes t}}{U(1)_K^t} + \text{monopole superpotentials} \right) , \tag{3.39}$$

$$\text{with mixed CS level : } K = 2 \begin{pmatrix} 1 & 1 & 1 & \cdots & 1 & 1 \\ 1 & 2 & 2 & \cdots & 2 & 2 \\ 1 & 2 & 3 & \cdots & 3 & 3 \\ \vdots & \vdots & \vdots & \ddots & \vdots & \vdots \\ 1 & 2 & 3 & \cdots & t-1 & t-1 \\ 1 & 2 & 3 & \cdots & t-1 & t \end{pmatrix} , \tag{3.40}$$

Here $\mathcal{T}_\Delta$ is a free theory of chiral multiplet with background CS level $-1/2$ for the $U(1)$ flavor symmetry [24] .

We now claim that the two descriptions for $\mathcal{T}_{(2,2t+3)}$ are actually equivalent.[5] One can check the following duality:

$$\mathcal{T}_{(2,2t+3)}^{GKS} \otimes \text{TFT}[(2, t+2)] \simeq D(\vec{k} = (2, t+2)) = \frac{T[SU(2)]}{(SU(2)_L)_2 \times (SU(2)_R)_{t+2}} , \tag{3.41}$$

from various BPS partition function computations. For example, the superconformal index for $D(\vec{k})$ can be computed using (A.17) and we find that

$$\mathcal{I}_{D(\vec{k}=(-2,-3))}^{\text{sci}}(\eta, \nu = 0) = \mathcal{I}_{\mathcal{T}_{(2,5)}^{GKS}}^{\text{sci}}(q, \eta, \nu = 0)$$
$$= 1 - q + \left( -\eta - \frac{1}{\eta} \right) q^{3/2} - 2q^2 + \left( -\eta - \frac{1}{\eta} \right) q^{5/2} - 2q^3 + \left( -\eta - \frac{1}{\eta} \right) q^{7/2} - 2q^4 + \dots$$
$$\mathcal{I}_{D(\vec{k}=(-2,-4))}^{\text{sci}}(\eta, \nu = 0) = \mathcal{I}_{\mathcal{T}_{(2,7)}^{GKS}}^{\text{sci}}(q, \eta, \nu = 0)$$
$$= 1 - q + \left( -\eta - \frac{1}{\eta} \right) q^{3/2} - 2q^2 - \eta q^{5/2} + \left( \frac{1}{\eta^2} - 1 \right) q^3 + \left( \frac{1}{\eta} - \eta \right) q^{7/2} + \frac{q^4}{\eta^2} + \dots .$$
$$\tag{3.42}$$

---

[5]In [53], they also found a dual description of the $\mathcal{T}_{(2,2t+3)}^{GKS}$ theory, which is $T_{\text{full}}[(t+1,1),(1,1),(1,1)]$.

For the round 3-sphere partition function case, we have ($\simeq$ means equality up to an overall phase factor as defined in (A.2)):

$$
\begin{aligned}
\left(\mathcal{Z}_{b=1}^{\mathrm{con}} \text{ of } D(\vec{k}=(2,t+2))\right) &\simeq \frac{1}{\sqrt{(2t+3)}} \sin\left(\frac{\pi}{2t+3}\right) , \\
\left(\mathcal{Z}_{b=1}^{\mathrm{con}} \text{ of } \mathrm{TFT}[\vec{k}=(2,t+2)]\right) &\simeq \frac{1}{\sqrt{|\det \mathcal{K}|}} = \frac{1}{2} , \\
\left(\mathcal{Z}_{b=1}^{\mathrm{con}} \text{ of } \mathcal{T}_{(2,2t+3)}^{GKS}\right) &\simeq \frac{2}{\sqrt{(2t+3)}} \sin\left(\frac{\pi}{2t+3}\right) ,
\end{aligned}
\tag{3.43}
$$

which again supports the proposed duality.

## 4 Discussion and Future Directions

In this paper, we provide an explicit field theory description of the 3D theory $\mathcal{T}_{(P,Q)}$ dual to the Virasoro minimal model $M(P,Q)$. Interestingly, the bulk theory exhibits very distinct IR phases—either gapped or $\mathcal{N}=4$ rank-0 SCFT—depending on whether the RCFT is unitary or not. The main results are presented in (2.24) and (3.22).

**Boundary Condition**  In this paper, we compute various partition functions on closed 3-manifolds to test the bulk-boundary correspondence. To directly observe the boundary rational chiral algebra, one needs to consider the theory on an open manifold with an appropriate boundary condition [34, 56–58]. Identifying the proper boundary condition that supports the Virasoro minimal models would be an interesting direction for future research.

**Mirror RCFTs of Minimal Models**  In 3D rank-0 SCFTs, there are two choices of topological twistings: $A$ and $B$ twisting. Our theory $\mathcal{T}_{(P,Q)}$ is expected to support the Virasoro minimal model at the boundary under one of these topological twistings. The other choice of twisting generally supports a different rational chiral algebra at the boundary. Understanding the mirror dual rational chiral algebras of the Virasoro minimal models would be a valuable avenue for further study.

**Other Minimal Models**  Recently, 3D dual theories for some supersymmetric $\mathcal{N}=1$ minimal models have been proposed [20]. Extending our work to other classes of minimal models, including these supersymmetric cases, would be an intriguing direction for future research. Some progress in this direction will be reported in [59].

## Acknowledgments

We would like to thank Yuji Tachikawa for the useful discussion. The work of DG and HK is supported in part by the National Research Foundation of Korea grant NRF-2022R1C1C1011979. DG also acknowledges support by Creative-Pioneering Researchers Program through Seoul National University.

# A    BPS partition functions of $D(\vec{k})$

The $T[SU(2)]$ theory, which is a basic building block of the $D(\vec{k})$ theory, is a 3D $\mathcal{N} = 4$ SQED with $N_f = 2$ (see Table 6). In terms of an $\mathcal{N} = 2$ subalgebra, the theory possesses

| Chiral multiplet | $U(1)_{\text{gauge}}$ | $R_{\nu=0}$ | $A$ | $F$ |
|:---:|:---:|:---:|:---:|:---:|
| $(\Phi_1, \Phi_2)$ | $(+1, -1)$ | $\frac{1}{2}$ | $\frac{1}{2}$ | $(+1, -1)$ |
| $(\Phi_3, \Phi_4)$ | $(+1, -1)$ | $\frac{1}{2}$ | $\frac{1}{2}$ | $(-1, +1)$ |
| $\Phi_0$ | $0$ | $1$ | $-1$ | $0$ |

**Table 6**. Matter contents of the $T[SU(2)]$ theory in terms of $\mathcal{N} = 2$ chiral multiplets. $(\Phi_1, \Phi_2)$ and $(\Phi_3, \Phi_2)$ form two $\mathcal{N} = 4$ hypermultiplets transforming as **2** under the $SU(2)_L$ flavor symmetry, with their Cartan denoted by $F$, normalized as $F \in \mathbb{Z}$. The theory has a $U(1)$ topological symmetry associated with the $U(1)$ gauge field, which is enhanced to $SU(2)_R$ in the IR. $\mathcal{N} = 4$ vector multiplet contains a neutral chiral multiplet $\Phi_0$. The theory has a superpotential $\mathcal{W} \propto \Phi_1 \Phi_0 \Phi_2 - \Phi_3 \Phi_0 \Phi_4$.

$SU(2)_L \times SU(2)_R \times U(1)_A$ flavor symmetry that commutes with the subalgebra. Various SUSY parition functions of the $T[SU(2)]$ theory and its variants have been computed in various literatures. To ensure self-containment, we reproduce these computations here.

**Squashed 3-sphere partition function [30–32]**    The squashed 3-sphere partition function $\mathcal{Z}^{S_b^3}$ of the $S$-duality wall theory is

$$\mathcal{Z}^{S_b^3}_{T[SU(2)]}(M_1, M_2; M, \nu) = \int \frac{dZ}{\sqrt{2\pi\hbar}} \mathcal{I}^\hbar_{T[SU(2)]}(Z, M_1, M_2; W), \quad \text{where}$$

$$\mathcal{I}^\hbar_{T[SU(2)]} := \exp\left(\frac{Z^2 + M_1^2 + 2M_2 Z}{\hbar}\right) \psi_\hbar\left(-W + (i\pi + \frac{\hbar}{2})\right) \tag{A.1}$$

$$\times \prod_{\epsilon_1, \epsilon_2 \in \{\pm 1\}} \psi_\hbar\left(\epsilon_1 Z + \epsilon_2 M_1 + \frac{W}{2} + (\frac{i\pi}{2} + \frac{\hbar}{4})\right)\Bigg|_{W := M + (i\pi + \hbar/2)\nu},$$

We define $\hbar := 2\pi i b^2$, where the $b$ is the squashing parameter of the squashed 3-sphere $S_b^3$ in (2.12).

The partition function has following phase factor ambiguity

$$\exp\left(i\pi(b^2 + b^{-2})\mathbb{Q} + i\pi\mathbb{Q}\right), \tag{A.2}$$

which depends on the background CS levels for $R$-symmetry and flavor symmetries, decoupled invertible TQFT, 3-manifold framing and so forth. The partition function depends on following parameters:

$$\begin{aligned}
M_1/M_2 &: \text{(rescaled) real masses for the the Cartan } U(1)\text{s of } SU(2)_L/SU(2)_R, \\
(M, \nu) &: \text{((rescaled) real mass, R-symmetry mixing in (2.14)) for the } U(1)_A.
\end{aligned} \tag{A.3}$$

The rescaled real mass $M$ is $b \times$ (real mass) and the squashed 3-sphere partition has $b \leftrightarrow b^{-1}$ symmetry when the (unrescaled) real masses and $\nu$ are fixed. The special function $\psi_\hbar$ in

the integrand is called quantum dilogarithm (Q.D.L) function, which is defined by [60]

$$\psi_\hbar(Z) := \begin{cases} \prod_{r=1}^{\infty} \frac{1-q^r e^{-Z}}{1-\widetilde{q}^{-r+1}e^{-\widetilde{Z}}} \, , & \text{if } |q| < 1 \\ \prod_{r=1}^{\infty} \frac{1-\widetilde{q}^r e^{-\widetilde{Z}}}{1-q^{-r+1}e^{-Z}} \, , & \text{if } |q| > 1 \end{cases} \tag{A.4}$$

with

$$q = e^{2\pi i b^2}, \quad \widetilde{q} := e^{2\pi i b^{-2}}, \quad \widetilde{Z} = Z/b^2 \, . \tag{A.5}$$

The $\psi_\hbar(Z)$ computes the squashed 3-sphere partition function of the $\mathcal{T}_\Delta$ theory [24], a massless free theory of single $\mathcal{N} = 2$ chiral multiplet with background CS level $-1/2$ for the $U(1)$ flavor symmetry. The $Z$ is $M + (i\pi + \frac{\hbar}{2}R(\Phi))$ with the rescaled real mass $M$ for the $U(1)$ flavor symmetry and the R-charge $R(\Phi)$ of the chiral field. The self-mirror property of the $T[SU(2)]$ theory implies that

$$\mathcal{Z}_{T[SU(2)]}^{S_b^3}(M_1, M_2; M, \nu) \simeq \mathcal{Z}_{T[SU(2)]}^{S_b^3}(M_2, M_1; -M, -\nu) \, . \tag{A.6}$$

Here $\simeq$ means the equality modulo a phase factor of the form in (A.2). Using the $T[SU(2)]$ partition function, the squashed 3-sphere partition function of $D(\vec{k})$ in Figure 3 can be computed as follows

$$\mathcal{Z}_{D(\vec{k})}^{S_b^3}(M, \nu)$$
$$= \int \left( \prod_{I=1}^{\sharp} \frac{\Delta(M_I) dM_I}{\sqrt{2\pi\hbar}} \exp\left( \frac{k^{(I)} M_I^2}{\hbar} \right) \right) \left( \prod_{I=1}^{\sharp-1} \mathcal{Z}_{T[SU(2)]}^{S_b^3}(M_I, M_{I+1}; M, \nu) \right) \, . \tag{A.7}$$

The $\Delta(M)$ is the contribution from a $SU(2)$ vector multiplet,

$$\Delta(M) = 2\sinh(M)\sinh(2\pi i M/\hbar) \, . \tag{A.8}$$

The partition function for $D(\vec{k})$ drastically simplifies when $b = 1$ and $\nu = 0$. For $T[SU(2)]$ theory, the partition function becomes [14, 61]

$$\mathcal{Z}_{T[SU(2)]}^{S_{b=1}^3}(M_1, M_2; M = 0, \nu = 0) \simeq \frac{1}{2} \frac{\sin\left( \frac{M_1 M_2}{\pi} \right)}{\sinh(M_1)\sinh(M_2)} \, . \tag{A.9}$$

For $D(\vec{k})$ theory,

$$\mathcal{Z}_{D(\vec{k})}^{S_{b=1}^3}(M = 0, \nu = 0)$$
$$\simeq 2 \int \left( \prod_{I=1}^{\sharp} \frac{dM_I}{2\pi} \exp\left( \frac{k^{(I)} M_I^2}{2\pi i} \right) \right) \sinh(M_1)\sinh(M_\sharp) \prod_{I=1}^{\sharp-1} \sin\left( \frac{M_I M_{I+1}}{\pi} \right) \, , \tag{A.10}$$
$$\simeq \frac{1}{\sqrt{2^{\sharp-2}|p|}} \sin\left( \frac{\pi}{|p|} \right) \, .$$

The integral in the middle line is simply a sum of Gaussian integrals and can be easily evaluated to obtain the final answer. Interestingly, the final result can be expressed as a

very simple function of $p$, which is related to $\vec{k}$ as shown in (3.2). In the computation, we use the following identity:

$$\det \overline{\mathcal{K}}(\vec{k}) = |p| ,$$

$$\text{where } \overline{\mathcal{K}}_{IJ} := \begin{cases} 1, & |I-J| = 1 \\ k^{(I)}, & I = J \\ 0, & \text{otherwise} \end{cases} \qquad (I, J = 1, \dots, \sharp) \tag{A.11}$$

**Superconformal index [62–64]** The generalized superconformal index $\mathcal{I}^{\mathrm{sci}}_{T[SU(2)]}$ for the $T[SU(2)]$ theory is $(\tilde{\eta}^2 := \eta)$

$$
\begin{aligned}
&\mathcal{I}^{\mathrm{sci}}_{T[SU(2)]}(m_1, u_1, m_2, u_2; \eta, \nu) \\
&= \sum_m \oint_{|u|=1} \frac{du}{2\pi i u} u^{2(m+m_2)} u_2^{2m} u_1^{2m_1} \mathcal{I}_\Delta(0, -q^{1/2}\tilde{\eta}^{-2}) \\
&\quad \times \prod_{\epsilon_1, \epsilon_2 \in \{\pm 1\}} \mathcal{I}_\Delta(\epsilon_1 m + \epsilon_2 m_1, \tilde{\eta} u^{\epsilon_1} u_1^{\epsilon_2} (-q^{1/2})^{1/2})\Big|_{\tilde{\eta} \to (-\eta q^{1/2})^{\nu/2}} , \\
&= \sum_m \oint_{|u|=1} \frac{du}{2\pi i u} (-q^{1/2})^{m+m_2} u^{2(m+m_2)} u_1^{2m_1} u_2^{2m} \mathcal{I}_\Delta(0, -q^{1/2}\tilde{\eta}^{-2}) \mathcal{I}_\Delta(m + m_1, -q^{1/2}\tilde{\eta} u u_1) \\
&\quad \mathcal{I}_\Delta(m - m_1, -q^{1/2}\tilde{\eta} u u_1^{-1}) \mathcal{I}_\Delta(-m + m_1, \tilde{\eta} u^{-1} u_1) \mathcal{I}_\Delta(-m - m_1, \tilde{\eta} u^{-1} u_1^{-1})\Big|_{\tilde{\eta} \to (-\eta q^{1/2})^{\nu/2}} .
\end{aligned}
\tag{A.12}
$$

In the middle, we changed the integral variable $u$ to $u(-q^{1/2})^{1/2}$, which corresponds to adjusting the mixing between the $U(1)$ R-symmetry and the gauge $U(1)$ symmetry, i.e.,

$$R_\nu \to \tilde{R}_\nu := R_\nu + \frac{1}{2} G , \tag{A.13}$$

where $G$ is the gauge charge. Since the index counts gauge-invariant operators, it remains unaffected by the mixing. In practice, the last expression is much easier to handle using Mathematica. The index depends on the following parameters:

$(m_1, u_1)/(m_2, u_2)$ : (monopole flux, fugacity) for the the Cartans of $SU(2)_L/SU(2)_R$ ,

$(\eta, \nu)$ : fugacity and R-symmetry mixing parameter for the $U(1)_A$ symmetry .

(A.14)

Here the tetrahedron index $\mathcal{I}_\Delta(m, u)$ is defined as [64]

$$
\mathcal{I}_\Delta(m, u) := \prod_{r=0}^{\infty} \frac{1 - q^{r - \frac{1}{2}m + 1} u^{-1}}{1 - q^{r - \frac{1}{2}m} u} = \sum_{e \in \mathbb{Z}} \mathcal{I}^c_\Delta(m, e) u^e,
$$

$$
\text{where } \mathcal{I}^c_\Delta(m, e) = \sum_{n = \lfloor e \rfloor}^{\infty} \frac{(-1)^n q^{\frac{1}{2}n(n+1) - (n + \frac{1}{2}e)m}}{(q)_n (q)_{n+e}} .
\tag{A.15}
$$

It computes the generalized superconformal index of the $\mathcal{T}_\Delta$ theory with the R-charge choice $R(\Phi) = 0$ where $(m, u)$ are (background monopole flux, fugacity) for the $U(1)$ flavor

symmetry. At general $R$-charge choice, the index becomes $\mathcal{I}_\Delta(m, u(-q^{1/2})^{R(\Phi)})$. As a consistency check for the formula, one can confirm the following self-mirror property in $q$-expansion

$$\mathcal{I}^{\mathrm{sci}}_{T[SU(2)]}(m_1, u_1, m_2, u_2; \eta, \nu) = \mathcal{I}^{\mathrm{sci}}_{T[SU(2)]}(m_2, u_2, m_1, u_1; \eta^{-1}, -\nu) . \tag{A.16}$$

Using the $T[SU(2)]$ index, the superconformal index of the $D(\vec{k})$ theory can be computed as follows:

$$\mathcal{I}^{\mathrm{sci}}_{D(\vec{k})}(\eta, \nu)$$
$$= \sum_{m_1, \ldots, m_m \in \mathbb{Z}_{\geq 0}} \oint \prod_{I=1}^{\sharp} \left( \frac{du_I}{2\pi i u_I} \Delta(m_I, u_I) u_I^{2k^{(I)} m_I} \right) \left( \prod_{I=1}^{\sharp-1} \mathcal{I}^{\mathrm{sci}}_{T[SU(2)]}(m_I, u_I, m_{I+1}, u_{I+1}; \eta, \nu) \right) . \tag{A.17}$$

Here $\Delta(m, u)$ is the contribution from a $SU(2)$ vector multiplet

$$\Delta(m, u) = \frac{1}{\mathrm{Sym}(m)}(q^{m/2}u - q^{-m/2}u^{-1})(q^{m/2}u^{-1} - q^{-m/2}u)$$
$$\text{with } \mathrm{Sym}(m) := \begin{cases} 2, & m = 0 \\ 1, & m \neq 0 \end{cases} . \tag{A.18}$$

One can also check the following

$$\mathcal{I}^{\mathrm{sci}}_{T[SU(2)]/(SU(2)_R)_{\pm 1}}(m_1, u_1; \eta, \nu)$$
$$= \sum_{m_2 \in \mathbb{Z}_{\geq 0}} \oint \frac{du_2}{2\pi i u_2} \Delta(m_2, u_2) u_2^{\pm 2m_2} \mathcal{I}^{\mathrm{sci}}_{T[SU(2)]}(m_1, u_1, m_2, u_2; \eta, \nu) = u_1^{\mp 2m_1} . \tag{A.19}$$

It provides a non-trivial check for the IR duality corresponding to the first move in Figure 4.

**Twisted partition functions [26–29]** Now let us compute the twisted partition functions $\mathcal{Z}^{\mathcal{M}_{g,p}}$ (2.11) of the $D(\vec{k})$ theory. For the computation, we first consider the integrand

of the squashed 3-sphere partition function (A.7) in an asymptotic limit $\hbar \to 0$ [65]:

$$\log \mathcal{I}^{\hbar}_{D(\vec{k})} \xrightarrow{\hbar \to 0} \frac{1}{\hbar}\mathcal{W}^{(\vec{k})}_0 + \mathcal{W}^{(\vec{k})}_1 + \dots ,$$

$$\mathcal{W}^{(\vec{k})}_0(\vec{Z}, \vec{M}; M, \nu) = \left(\sum_{I=1}^{\sharp} (\pm 2\pi i M_I + k^{(I)} M_I^2)\right) + \sum_{I=1}^{\sharp-1} \mathcal{W}^{T[SU(2)]}_0(Z_I, M_I, M_{I+1}; M, \nu) ,$$

$$\text{where } \mathcal{W}^{T[SU(2)]}_0(Z, M_1, M_2; M, \nu) = Z^2 + M_1^2 + 2M_2 Z + \text{Li}_2(e^{M+i\pi\nu})$$
$$+ \sum_{\epsilon_1, \epsilon_2 \in \{\pm 1\}} \text{Li}_2\left(-e^{-\epsilon_1 Z - \epsilon_2 M_1 - \frac{(M+i\pi(\nu-1))}{2}}\right) ,$$

$$\mathcal{W}^{(\vec{k})}_1(\vec{Z}, \vec{M}; M, \nu) = \sum_{I=1}^{\sharp} \log(\sinh(M_I)) + \sum_{I=1}^{\sharp-1} \mathcal{W}^{T[SU(2)]}_1(Z_I, M_I, M_{I+1}; M, \nu) \text{ with}$$

$$\mathcal{W}^{T[SU(2)]}_1(Z, M_1, M_2) = -\frac{\nu}{2}\log(1 + e^{M+i\pi\nu})$$
$$+ \frac{\nu-1}{4} \sum_{\epsilon_1, \epsilon_2 \in \{\pm 1\}} \log\left(1 + e^{-\epsilon_1 Z - \epsilon_2 M_1 - \frac{(M+i\pi(\nu-1))}{2}}\right) .$$

$$(A.20)$$

We use the following asymptotic behavior of Q.D.L,

$$\log \psi_\hbar(Z) \xrightarrow{\hbar \to 0} \frac{\text{Li}_2(e^{-Z})}{\hbar} - \frac{1}{2}\log(1 - e^{-Z}) + \dots . \qquad (A.21)$$

Then, the Bethe-vacua of the $D(\vec{k})$ theory are obtained as follows ($I = 1, \dots, (\sharp - 1)$ while $J = 1, \dots, \sharp$)

$$\mathcal{S}^{\text{BE}}_{D(\vec{k})}(M, \nu) = \left\{\vec{z}, \vec{m} : \exp(\partial_{Z_I}\mathcal{W}^{(\vec{k})}_0)\big|_* = \exp(\partial_{M_J}\mathcal{W}^{(\vec{k})}_0)\big|_* = 1, \, m_J^2 \neq 1\right\}/\mathbf{W} ,$$
$$\text{with } * : Z_I \to \log z_I, \, M_J \to \log m_J . \qquad (A.22)$$

Here $\mathbf{W}$ denotes the Weyl subgroup $\mathbb{Z}_2^{\sharp}$ of the $SU(2)^{\sharp}$ gauge symmetry, which acts on the Bethe-vacua as

$$\mathbf{W} : \, m_J \to 1/m_J \text{ for each } J = 1, \dots, \sharp . \qquad (A.23)$$

Handle gluing $\mathcal{H}$ and fibering operator $\mathcal{F}$ of the $D(\vec{k})$ theory are ($\vec{X} := (\vec{Z}, \vec{M})$)

$$\mathcal{H}_{D(\vec{k})}(\vec{z}, \vec{m}; M, \nu) = \frac{e^{i\delta}}{|\mathbf{W}|^2}\left(\det_{A,B}\left(\partial_{X_A}\partial_{X_B}\mathcal{W}^{(\vec{k})}_0\right)\right)\exp\left(-2\mathcal{W}^{(\vec{k})}_1\right)\Bigg|_* ,$$

$$\mathcal{F}_{D(\vec{k})}(\vec{z}, \vec{m}; M, \nu) = \exp\left(-\frac{\mathcal{W}^{(\vec{k})}_0 - \vec{X} \cdot \partial_{\vec{X}}\mathcal{W}^{(\vec{k})}_0 - M\partial_M\mathcal{W}^{(\vec{k})}_0}{2\pi i}\right)\Bigg|_* . \qquad (A.24)$$

Here $|\mathbf{W}| = 2^{\sharp}$ and $e^{i\delta}$ is a phase factor which is sensitive to the subtle overall phase factor in (A.2). We fix the phase ambiguity by requiring that

$$\mathcal{Z}^{\mathcal{M}_{g=0,p=0}}_{D(\vec{k})}(M = 0, \nu = \pm 1) = \sum_{(\vec{z}, \vec{m}) \in \mathcal{S}_{\text{BE}}} \mathcal{H}^{-1}_{D(\vec{k})} = 1 . \qquad (A.25)$$

For $D(\vec{k})$ theory with $\vec{k} = (k_1, k_2)$, the twisted partition functions are studied in [14]. There are $2 \times (|k_1 k_2 - 1| - 1)$ Bethe-vacua whose handle gluing operators

$$\{\mathcal{H}_{D(\vec{k}=(k_1,k_2))}(\vec{z}, \vec{m}) \; : \; (\vec{z}, \vec{m}) \in \mathcal{S}_{\mathrm{BE}}(M = 0, \nu = \pm 1)\} = \left\{ \frac{|k_1 k_2 - 1|}{\sin^2\left(\frac{\pi n}{k_1 k_2 - 1}\right)}^{\otimes 2} \right\}_{n=1}^{|k_1 k_2 - 1| - 1} .$$

(A.26)

Further, one can check that the set $\mathbf{HF} := \{(\mathcal{H}^{-1/2}, \mathcal{F}) \; : \; (\vec{z}, \vec{m}) \in \mathcal{S}_{\mathrm{BE}}\}$ of $D(\vec{k} = (k_1, k_2))$ has following factorization properties

$$\mathbf{HF} = \widetilde{\mathbf{HF}} \times \begin{cases} \{(\frac{1}{2}, 1), (\frac{1}{2}, 1), (\frac{1}{2}, 1), (\frac{1}{2}, -1)\}, & k_1 \text{ and } k_2 \text{ are both even}, \\ \{(\frac{1}{2}, 1), (\frac{1}{2}, 1), (\frac{1}{2}, i), (\frac{1}{2}, i^{-1})\}, & \text{one of them is odd} \\ \{(\frac{1}{\sqrt{2}}, 1), (\frac{1}{\sqrt{2}}, i)\} \text{ or } \{(\frac{1}{\sqrt{2}}, 1), (\frac{1}{\sqrt{2}}, i^{-1})\}, & \text{both are odd} \end{cases}$$

(A.27)

We will understand the above factorization pattern by analyzing the decoupled TQFT $\mathrm{TFT}[\vec{k}]$ in the next section, .

# B  Decoupled TQFT $\mathbf{TFT}(\vec{k})$

From the 't Hooft anomaly (3.9) of the $(\mathbb{Z}_2)^\sharp$ 1-form symmetry in $D(\vec{k})$, we expect that the $D(\vec{k})$ theory in the IR contains a decoupled topological field theory which has the same anomaly. The simplest choice is the $U(1)_{\mathcal{K}}^\sharp$ theory with the mixed CS term $\mathcal{K}$ of the following form:

$$\frac{\mathcal{K}}{2} = (\overline{\mathcal{K}} \text{ in (A.11)}) \; (\mathrm{mod} \; 2) .$$

(B.1)

This implies that

$$\det(\mathcal{K}/2) = \det(\overline{\mathcal{K}}) = |p| \; (\mathrm{mod} \; 2) ,$$
$$\Rightarrow |\det(\mathcal{K})| = \begin{cases} \in 2^\sharp \times (2\mathbb{Z}_{\geq 0} + 1), & \text{odd } p \\ \in 2^\sharp \times (2\mathbb{Z}_{\geq 0}), & \text{even } p \end{cases}$$

(B.2)

On the other hand, for the $U(1)_{\mathcal{K}}^\sharp$ theory (when $\mathcal{K}$ is non-degenerate)

$$\det(\mathcal{K}) = (\text{the number of ground states on } \mathbb{T}^2, \text{ i.e. Bethe-vacua})$$
$$\leq 2^\sharp .$$

(B.3)

since all the Bethe-vacua of the decoupled TQFT are connected to each other by an action of the $(\mathbb{Z}_2)^\sharp$ 1-form symmetry. Combining (B.2) and (B.3), the $\det \mathcal{K}$ is determined as follows:

$$\det(\mathcal{K}) = \begin{cases} 2^\sharp, & \text{odd } p \\ 0, & \text{even } p \end{cases}$$

(B.4)

$\sharp = 2$ **and** $k_1, k_2 \in 2\mathbb{Z}$   In the case, $p$ is always odd and the possible candidates for $\mathcal{K}$ are ($n \in \mathbb{Z}$)

$$\mathcal{K} = \begin{pmatrix} 4n & \pm 2 \\ \pm 2 & 0 \end{pmatrix} \text{ or } \begin{pmatrix} 0 & \pm 2 \\ \pm 2 & 4n \end{pmatrix} \tag{B.5}$$

But all the $\mathcal{K}$s are actually equivalent to

$$\mathcal{K} \simeq \begin{pmatrix} 0 & 2 \\ 2 & 0 \end{pmatrix} . \tag{B.6}$$

Here $\mathcal{K} \simeq \mathcal{K}'$ is an equivalence between $U(1)_{\mathcal{K}}^{\sharp}$ and $U(1)_{\mathcal{K}'}^{\sharp}$ up to a following redefinition of gauge fields $\vec{A}$

$$\vec{A} \to M \cdot \vec{A} \text{ with } M \in GL(2\sharp, \mathbb{Z}) . \tag{B.7}$$

We choose $GL(2\sharp, \mathbb{Z})$ instead of $GL(2\sharp, \mathbb{R})$ to preserve the monopole charge quantization. More explicitly, the equivalence relation is

$$\mathcal{K} \simeq \mathcal{K}' \text{ if } \mathcal{K}' = M^T \mathcal{K} M \text{ with a } M \in GL(2\sharp, \mathbb{Z}) . \tag{B.8}$$

So, in the case, the decoupled TQFT is uniquely determined

$$\text{TFT}[(k_1, k_2)] \simeq U(1)_{\mathcal{K}}^2 \text{ with } \mathcal{K} = \begin{pmatrix} 0 & 2 \\ 2 & 0 \end{pmatrix} , \tag{B.9}$$

which is the toric-code TQFT. It has the set of $\mathbf{HF} := \{|S_{0\alpha}|, \exp(2\pi i h_\alpha)\}$ as follows [6]

$$(\mathbf{HF} \text{ of TFT}[(k_1, k_2)]) = \left\{ (\frac{1}{2}, 1)^{\otimes 3}, (\frac{1}{2}, -1) \right\} , \tag{B.10}$$

which explains the corresponding factorization property in (A.27).

$\sharp = 2$ **and** $k_1 \neq k_2 \pmod 2$   In the case, $p$ is always odd and there is only one consistent choice of $\mathcal{K}$ up to the equivalence, which is

$$\text{TFT}[(k_1, k_2)] \simeq U(1)_{\mathcal{K}}^2 \text{ with } \mathcal{K} = \begin{pmatrix} 2 & 2 \\ 2 & 0 \end{pmatrix} , \tag{B.11}$$

whose $\mathbf{HF} := \{|S_{0\alpha}|, \exp(2\pi i h_\alpha)\}$ is

$$(\mathbf{HF} \text{ of TFT}[(k_1, k_2)]) = \left\{ (\frac{1}{2}, 1)^{\otimes 2}, (\frac{1}{2}, i), (\frac{1}{2}, \frac{1}{i}) \right\} , \tag{B.12}$$

which explains the corresponding factorization property in (A.27).

---

[6] For the $U(1)_{\mathcal{K}}^{\sharp}$ theory with non-degenerate $\mathcal{K}$, there are $\det \mathcal{K}$ simple objects (or Bethe-vacua), $\alpha = 0, \ldots, \det \mathcal{K} - 1$, which are solutions of the Bethe equations, $\mathcal{S}_{\text{BE}} := \{\vec{z} : \prod_{I=1}^{\sharp} z_I^{\mathcal{K}_{IJ}} = 1, \text{ for } J = 1, \ldots, \sharp\}$. The $S_{0\alpha}$ of Bethe-vacuum $\vec{z}_\alpha \in \mathcal{S}_{\text{BE}}$ is $1/\sqrt{|\det \mathcal{K}|}$ for all $\alpha$ and the $e^{2\pi i h_\alpha}$ is given by the fibering operator $\mathcal{F} = \exp(\frac{\sum_{I,J} \mathcal{K}_{IJ} Z_I Z_J}{4\pi i})$ with $\vec{Z} = \log \vec{z}_\alpha$.

$\sharp = 2$ **and** $k_1, k_2 \in 2\mathbb{Z} + 1$    In the case, $p$ is always even and the possible candidates for the $\mathcal{K}$ are

$$\mathcal{K} \simeq 2 \begin{pmatrix} a & a \\ a & a \end{pmatrix} \simeq 2 \begin{pmatrix} 0 & 0 \\ 0 & a \end{pmatrix} \text{ with } a \in 2\mathbb{Z} + 1 . \tag{B.13}$$

Ignoring the 1st gauge field $A_1$ which does not appear in the action, the decoupled TQFT is nothing but the $U(1)_{2a}$ theory. From the constraint in (B.3), which is $|2a| \leq 2^2$ for our case, the possible values of $a$ are $\pm 1$. Thus, the decoupled TQFT is $U(1)_{\pm 2}$ theory whose $\mathbf{HF} := \{|S_{0\alpha}|, \exp(2\pi i h_\alpha)\}$ is

$$(\mathbf{HF} \text{ of TFT}[(k_1, k_2)]) = \left\{ (\frac{1}{\sqrt{2}}, 1), (\frac{1}{\sqrt{2}}, i^{\pm 1}) \right\} , \tag{B.14}$$

which explains the corresponding factorization property in (A.27).

For higher values of $\sharp \geq 3$, the analysis becomes more complicated, and we could not uniquely determine $\mathcal{K}$. From the factorization properties of the set $\mathbf{HF}$ for several $D(\vec{k})$s, we observe that

$$(\text{the number of Bethe-vacua of the decoupled TQFT}) = \begin{cases} 2^\sharp, & \text{odd } p \\ 2^{\sharp-1}, & \text{even } p \end{cases} \tag{B.15}$$

For odd $p$, this is compatible with (B.3) and (B.4). For abelian CS theory, it is generally true that

$$(\text{3-sphere partition function}) = 1/\sqrt{(\text{the number of Bethe-vacua})} . \tag{B.16}$$

Thus, from (B.15), we have

$$|(\mathcal{Z}^{\text{con}} \text{ of TFT}(\vec{k}))| = \begin{cases} 2^{-\sharp/2}, & p \in 2\mathbb{Z} + 1 \\ 2^{(1-\sharp)/2}, & p \in 2\mathbb{Z} \end{cases} \tag{B.17}$$

Combined with (A.10), we obtain

$$|(\mathcal{Z}^{\text{con}} \text{ of } \mathcal{D}(p, q))| = \frac{|(\mathcal{Z}^{\text{con}} \text{ of } D(\vec{k}))|}{|(\mathcal{Z}^{\text{con}} \text{ of TFT}(\vec{k}))|} = \begin{cases} \frac{2}{\sqrt{|p|}} \sin\left(\frac{\pi}{|p|}\right), & p \in 2\mathbb{Z} + 1 \\ \sqrt{\frac{2}{|p|}} \sin\left(\frac{\pi}{|p|}\right), & p \in 2\mathbb{Z} \end{cases} \tag{B.18}$$

Note that the partition function is independent on the choice of $\vec{k}$ as expected from $i$) in (3.11).

For even $p$, a $\mathbb{Z}_2$ subgroup of the UV $(\mathbb{Z}_2)^\sharp$ 1-form symmetry is absent in the decoupled TFT$(\vec{k})$, which captures the anomaly of the 1-form symmetry. This implies that

$$\text{For } p \in 2\mathbb{Z}, \quad \mathcal{D}(p, q) \text{ has a non-anomalous } \mathbb{Z}_2 \text{ 1-form symmetry}, \tag{B.19}$$

which can be identified with the absent $\mathbb{Z}_2$ subgroup .

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
