# Peer review of "Non-hyperbolic 3-manifolds and 3D field theories for 2D Virasoro minimal models"

_SciPost Physics_

## Round 2 · Referee Report · Anonymous (Referee 1) · 2025-9-17

Report

This is a continuation of the authors' series of works on the 3d/3d correspondence on non-hyperbolic 3-manifolds and 3d (possibly-non-unitary) TQFTs. The main new result in this paper is the explicit identification of the 3-manifolds for the bulk TQFTs for the Virasoro minimal models, including the non-unitary ones, as a specific class of Seifert fibrations over $S^2$ with three singular fibers.

As always in this series of papers, the content is mostly exploratory. But the paper comes with a nice summary of all the basic formulas, concepts and references needed to follow the computations, so that any interested reader should be able to reproduce the computation and confirm that the claimed correspondences do hold.

That said, before being able to be published, the referee thinks that the paper has to have a more nontrivial check in the non-unitary case. The reason behind this request is the following. The TQFT for the Virasoro minimal model with the label $(P,Q)$ is more or less the product of $SU(2)_{P-2}$ and $SU(2)_{Q-2}$. And in the unitary case $Q=P+1$, the agreement (the paper's (3.24)) was in terms of $D(P,P-R)$ being equal to $SU(2)_{P-2}$ and $D(Q,S)$ being equal to $SU(2)_{Q-2}$. More generally, the authors' proposal (3.22) is more or less a product of $D(P,P-R)$ and $D(Q,S)$. But in all the non-unitary examples provided in the paper, one of $D(P,P-R)$ and $D(Q,S)$ was always trivial. The referee thinks it necessary that the authors check at least one case where both $D(P,P-R)$ and $D(Q,S)$ are nontrivial and do not satisfy $q=\pm1$ mod $p$.

Furthermore, if the authors' proposal is correct in general, would it not mean that $D(P,P-R)$ is (more or less) $SU(2)_{p-2}$ and $D(Q,S)$ is (more or less) $SU(2)_{Q-2}$, indepent of $R$ or $S$? These points need to be clarified.

THere are also many trivial typos and minor notational inconsistencies. It can be said that those trivial typos won't affect the understandability of the paper, but still the referee thinks these should be eliminated as much as possible. Some examples are listed below:

  • Some lines above (2.13), the data $(M,\nu)$ is introduced. There is no explanation on this $M$ or $\nu$ at this point. What makes this worse is that the same symbol $M$ (with the same font) has been used to denote the 3-manifold in the same page.

  • Some lines above (2.18), the all flat connections -> all the flat connections

  • One line below (2.33), the So the irreducible -> So the irreducible ...

  • One line above (3.17), slop $p$ -> slope $p$

  • In the second line of (3.22), there is no closing parenthesis for the opening parenthesis (.

  • In the paragraph above (3.24), "resp. Q" should be written as "resp.~Q" in the TeX, otherwise TeX automatically introduces two spaces after a period following a lower-case alphabet.

  • Below (A.10), the final answer.Interestingly -> the final answer. Interestingly (A space was missing)

  • Above (A.18), a SU(2) -> an SU(2)

  • Within (B.8), a M -> an M

Recommendation

Ask for major revision

---

## Round 2 · Referee Report · Anonymous (Referee 2) · 2025-9-26

Strengths

1) The paper found connection between theories that have only been explored recently and well known models in the literature. 2) The paper is very clear and well written. 3) Concrete examples backed by explicit computations of superconformal indices were given.

Weaknesses

None

Report

The paper makes an interesting connection between certain 3D bulk models and the well known 2D Virasoro-minimal models via bulk-boundary correspondence. To achieve this, they utilized 3D-3D correspondence between Seifert fiber spaces and 3D gapped theories or 3D N=4 rank 0 theories (which recently gained much attention in the literature). This allows the authors to find a subset of 3D theories T_(P,Q) that are dual to both unitary and non-unitary 2D minimal models.

The Virasoro-minimal models includes many important theories such as Yang-Lee theory and therefore making it a very interesting set of results to find their 3D bulk duals. The authors made clear arguments for their conjectured T_(P,Q) bulk duals such as providing dictionaries between conformal primaries of the minimal models and the irreducible characters in the bulk theory. In some examples, superconformal indices were explicitly computed to check the result. These are interesting results that can lead to many extensions in the future and can unveil deeper understanding of these 2D minimal models. Thus, I recommend this paper to be published in its current form.

I have one question: For Figure 2, and equation (3.4) an interesting relation was drawn between the T_irred theory and the T_full theory. The T_full theory is related to the S1 reduction of the well known 4d N=2 class S theories. Is this relation just a coincidence with the T_2 theory or more general T_N theories (or perhaps more general class S theories) also plays a role in finding T_(P,Q) theories.

Requested changes

I believe the paper is good to publish in its present form.

Recommendation

Publish (meets expectations and criteria for this Journal)

---

## Round 2 · Referee Report · Anonymous (Referee 3) · 2025-10-23

Report

In the manuscript, the authors provide a family of Seifert 3-manifolds that correspond to 2-dimensional Virasoro minimal models. Physically, the correspondence hypothetically works in the following way. A twisted compactification of 6d (2,0) $A_1$ SCFT on the 3-manifolds gives an effective 3d supersymmeric theory, which, in the IR flows to either a 3d TQFT or a 3d $\mathcal{N}=4$ SCFT. The Virasoro minimal model then describes the 2d chiral CFT on the boundary of the unitary 3d TQFT, in the first case, and the non-unitary one obtained by topological twisting in the second case. In practice, the correspondence is established by matching various quantities on the 2d CFT side (such as conformal dimensions of primaries or quantum dimensions of the topological lines) with certain topological invariants of the 3-manifold (such as Chern-Simons invariants or Reidemeister-Turaev torsion of flat connections). This work can be considered, in a sense, as a continuation of the previous work about correspondence between Seifert-manifolds and 3d TQFTs, with some overlap of authors. 

I find that the manuscript is in general well-written, with only minor issues that l list below. I believe that the results will be interesting for mathematical physicist working on related topics, in particular on 3d topological and supersymmetric quantum field theories. I would like to recommend it for publication.

Requested changes

1) I find the discussion on page 4-6 a little confusing. On one hand, it looks like there is a free choice of topological twist: A or B, which is supposed to be the "value" of the 'top' subscript, and many quantities depend on it. On the other hand, the untwisted supersymmetric 3d theory $\mathcal{T}(P,Q)$, the boundary chiral CFT, $\chi M(P,Q)$, as well as various Chern-Simons theory quantities seem to be fixed (i.e. do not have 'top' dependence). The authors should make it more clear if there is always some particular (unknown to the authors?) choice of A or B is assumed.  The authors do make some relevant comments in Section 4, but I think this point should be clarified early in the paper.

2) The notation $M$ is used both for the 3-manifold and for the mass, it should be different.

3) I find the statement in (3.5) a little strange. I suggest that the authors either write what that topological theory is or explicitly state that they do not know (and that it is not important for other analysis in the manuscript). 

4) Some misprints that I noticed: 

  • "prtition", bottom of page 4
  • "m" after $H_1(\ldots)" in the sentence below (3.1) on page 11
  • "which is expected to a", bottom of page 11
  • "\otimes" instead of "$\times$" in (3.7)
  • "expec" above (3.16)
  • "slop" above (3.17)

Recommendation

Ask for minor revision

---

## Editorial Decision

awaiting_resubmission